# Distinct pathways for evolution of enhanced receptor binding and cell entry in SARS-like bat coronaviruses

**Alexandra L. Tse**[1‡], **Cory M. Acreman**[2‡], **Inna Ricardo-Lax**[3], **Jacob Berrigan**[1], **Gorka Lasso**[1], **Toheeb Balogun**[4], **Fiona L. Kearns**[4], **Lorenzo Casalino**[4], **Georgia L. McClain**[3], **Amartya Mudry Chandran**[1], **Charlotte Lemeunier**[1], **Rommie E. Amaro**[4], **Charles M. Rice**[3], **Rohit K. Jangra**[1,5], **Jason S. McLellan**[2*], **Kartik Chandran**[1*], **Emily Happy Miller**[1,6*]

1 Department of Microbiology & Immunology, Albert Einstein College of Medicine, Bronx, New York, New York, United States of America, 2 Department of Molecular Biosciences, The University of Texas at Austin, Austin, Texas, United States of America, 3 Laboratory of Virology and Infectious Disease, The Rockefeller University, New York, New York, United States of America, 4 Department of Molecular Biology, University of California San Diego, La Jolla, California, United States of America, 5 Present address: Department of Microbiology and Immunology, Louisiana State University Health Sciences Center-Shreveport, Shreveport, Louisiana, United States of America, 6 Department of Medicine, Albert Einstein College of Medicine, Bronx, New York, New York, United States of America

‡ These authors share first authorship on this work.
* jmclellan@austin.utexas.edu (JSM); kartik.chandran@einsteinmed.edu (KC); emily.miller@einsteinmed.edu (EHM)

**Data Availability Statement:** The raw data generated in this study has been deposited in the Figshare database under accession code https://doi.org/10.6084/m9.figshare.27107101. Images of

## Abstract

Understanding the zoonotic risks posed by bat coronaviruses (CoVs) is critical for pandemic preparedness. Herein, we generated recombinant vesicular stomatitis viruses (rVSVs) bearing spikes from divergent bat CoVs to investigate their cell entry mechanisms. Unexpectedly, the successful recovery of rVSVs bearing the spike from SHC014-CoV, a SARS-like bat CoV, was associated with the acquisition of a novel substitution in the S2 fusion peptide-proximal region (FPPR). This substitution enhanced viral entry in both VSV and coronavirus contexts by increasing the availability of the spike receptor-binding domain to recognize its cellular receptor, ACE2. A second substitution in the S1 N–terminal domain, uncovered through the rescue and serial passage of a virus bearing the FPPR substitution, further enhanced spike:ACE2 interaction and viral entry. Our findings identify genetic pathways for adaptation by bat CoVs during spillover and host-to-host transmission, fitness trade-offs inherent to these pathways, and potential Achilles' heels that could be targeted with countermeasures.

## Author summary

The recent emergence of several highly virulent human coronaviruses, SARS-CoV, MERS-CoV and SARS-CoV-2, underscores the risk coronaviruses can pose to the human population. Bat coronaviruses (CoVs) are of particular concern due to their potential to

rVSV rescue in this study have been deposited in the Figshare database under accession codes https://doi.org/10.6084/m9.figshare.26999251 and https://doi.org/10.6084/m9.figshare.27106333. Nanopore sequencing data of rVSV-SHC014-CoV (A835D) and rVSV-SHC014-CoV (F294L+A835D +T842A) are available as assembly files in S1 File and S2 File (plaintext file in FASTA format) and raw reads are deposited in the Sequence Read Archive (SRA) under accession PRJNA1178633.

**Funding:** This work was supported by grant INV-031624 from the Bill & Melinda Gates Foundation (https://www.gatesfoundation.org/) (to K.C. and J. S.M) and the National Institutes of Health (https:// www.nih.gov/) (R01AI132633 to K.C., P01AI165075 to C.M.R). C.M.R. was additionally supported by the Meyer Foundation (https:// meyerfoundation.org/), the BAWD Foundation (https://www.causeiq.com/organizations/bawd-foundation,133922346/), The G. Harold and Leila Y. Mathers Charitable Foundation (https:// mathersfoundation.org/), and Fast Grants (www. fastgrants.org). J.S.M. was additionally supported by Welch Foundation (https://welch1.org/) grant number F-0003-19620604. AL.T. and J.B. were additionally supported by the NIH training grants T32-GM149364 (Medical Scientist Training Program) and T32AI070117 (Geographic Medicine and Emerging Infections), respectively, at Albert Einstein College of Medicine. E.H.M was additionally supported by the Institute for Clinical and Translational Research at Einstein and Montefiore (https://einsteinmed.edu/centers/ictr) (K12TR004411). I.R.L. was supported by a Bulgari Women & Science fellowship (https://www. rockefeller.edu/support-our-science/women-and-science/bulgari/). R.K.J. was partly supported by NIH grants (P20GM134974 and R21AI156482). T.B. is supported by the A.G. Leventis Foundation (https://www.leventisfoundation.org/), Shurl and Kay Curci Foundation (https://curcifoundation.org/), and Thermo Fisher Scientific Antibody Fellowships (https://www.thermofisher.com/us/en/home/life-science/antibodies/thermo-fisher-scientific-antibody-scholarship-program.html). The funders had no role in study design, data collection and analysis, decision to publish, or preparation of the manuscript.

**Competing interests:** K.C. is a member of the scientific advisory board and holds shares in Integrum Scientific, LLC. K.C. is a cofounder of and holds shares in Eitr Biologics Inc.

adapt to new hosts. Here, we attempted to generate recombinant vesicular stomatitis viruses (rVSVs) bearing the spike glycoproteins from several SARS-like bat CoVs to study their cell entry mechanisms. We identified two mutations in the SHC014-CoV spike that afforded successful recovery of an rVSV bearing this spike by greatly increasing viral entry. Interestingly, these mutations occur outside the receptor-binding domain (RBD) but enhance spike-receptor interaction nevertheless. These and other results herein establish that these mutations serve to "open" the spike and thereby augment virus-receptor engagement. Our work uncovers new genetic pathways that could contribute to the adaptation of bat CoVs during host spillover. However, these mutations also render the spike more susceptible to neutralizing antibodies that recognize the RBD, pointing to fitness tradeoffs associated with these pathways.

## Introduction

The recent emergence of multiple human coronaviruses—SARS-CoV, MERS-CoV, and SARS-CoV-2—accompanied by disease epidemics of regional or global scope, has highlighted the urgent need to identify related animal coronaviruses (CoVs), understand their biology and zoonotic potential, and pre-position countermeasures. Efforts to sample and sequence CoVs circulating in nature have identified a diverse, globally distributed group of viruses in bats [1–3]. Studies performed with authentic coronaviruses, pseudotyped viral vectors bearing bat-origin CoV spikes, and/or recombinant spike proteins have shown that many of these agents can enter and infect human cells, pointing to bats as major reservoirs for novel CoVs with the potential for zoonotic transmission to humans [4–14]. However, these findings have also demonstrated a continuum of cell entry efficiencies that could not be fully explained by differences in spike:receptor binding affinity alone, indicating the existence of additional entry barriers to human infection by some bat-origin CoVs [3,5,15–19].

As a case in point, a large sequencing study conducted in horseshoe bats collected from Yunnan Province, China, determined full-length genome sequences of seven CoVs belonging to the subgenus *Sarbecovirus*, genus *Betacoronavirus*, including two from novel agents—Rs3367 and RsSHC014 [4]. The authors also recorded the first successful isolation of a replication-competent SARS-like CoV (SL-CoV), WIV-1-CoV, that was almost identical to SL-CoV Rs3367 in sequence and demonstrated that it could replicate in human cells. However, they could not recover a virus corresponding to RsSHC014 (hereafter, SHC014-CoV) from bat fecal samples. Interestingly, although later studies showed that the receptor-binding domains (RBDs) of both WIV-1-CoV and SHC014-CoV spikes could recognize human angiotensin-converting enzyme-2 (ACE2)—the cell entry receptor for SARS-CoV, SARS-CoV-2, and many other sarbecoviruses—with high affinity [20,21], Menachery and colleagues reported that only the WIV-1-CoV spike could mediate high levels of lentiviral vector transduction into cells over-expressing human ACE2 [13]. Indeed, they measured little or no activity for the SHC014-CoV spike in this assay. Unexpectedly, however, authentic CoVs bearing the SHC014-CoV spike could be rescued by reverse genetics, replicated in human airway cultures, and were virulent in mice, leading the authors to conclude that, despite their results with pseudotyped viruses, the SHC014-CoV spike was 'poised' to mediate infections in humans [13]. Subsequent studies have incorporated SHC014-CoV spike pseudotypes into larger panels of single-cycle viruses for analyses of antibody-mediated neutralization but have not investigated their entry-related properties in detail [22–25], leaving open questions about potential molecular incompatibilities between this and other bat-origin CoV spikes and human cells.

The coronavirus spike glycoprotein, S, forms homotrimers embedded in the membrane envelope of the virion. The mature S protein comprises two subunits, S1 and S2, generated by post-translational cleavage of a precursor polypeptide. The receptor-binding subunit S1 is variable in sequence and associates closely with the more conserved membrane fusion subunit S2 [26]. Intersubunit interactions influence the conformational states and dynamics of both subunits, regulating exposure of the RBDs in S1, their engagement with cellular receptors, and subsequent entry-related rearrangements in S1 and S2 that drive membrane fusion between viral and cellular membranes [27].

A large body of evidence points to a critical role for RBD–receptor interactions in influencing viral host range. Although most host-range mutations in the RBD directly alter spike-receptor contacts, others—such as T372A in the RBD of SARS-CoV-2—also enhance receptor binding and viral fitness through conformational effects on the spike [17,28,29]. Further, recent findings with multiple coronaviruses indicate that sequences distal to the RBD–receptor interface, including those at the S1–S2 interface and/or in S2, can also impact viral multiplication, virulence, and host-to-host transmission [18,30–32]. One prominent example is D614G, a substitution near the S1–S2 boundary in SARS-CoV-2, which is hypothesized to disrupt a salt bridge with K854 in the S2 fusion peptide-proximal region (FPPR) [33]. This substitution rapidly increased in frequency and became fixed in the viral population early in the COVID-19 pandemic [34]. The enhanced capacity of mutant D614G spikes to mediate cell entry, with attendant advantages for intra-host viral multiplication and host-to-host transmission, has been proposed as the molecular basis of this selective sweep [35–37].

Recombinant vesicular stomatitis viruses (rVSVs) encoding coronavirus spike proteins as their only entry glycoprotein have proven useful as biosafety level-2 (BSL-2) surrogates to study viral entry and screen countermeasures. They also afford a safe setting for forward-genetic studies of structure-function relationships in these spikes. Herein, we attempted to generate rVSVs bearing spikes from SHC014-CoV and several other related bat-origin CoVs. Although some of these viruses could be readily recovered from plasmids and propagated in cell culture, others, including rVSV-SHC014-CoV S, were much more challenging to recover, in accordance with previously reported work with single-cycle pseudotypes. Unexpectedly, the successful rescue of rVSV-SHC014-CoV S was associated with the acquisition of a novel point mutation leading to an A835D substitution in the FPPR region of S2 that greatly enhanced entry in the context of both VSV and coronavirus particles. We also identified a second substitution in the N–terminal domain (NTD) of S1, F294L, that further increases spike:ACE2 interaction and viral entry. Our findings reveal molecular features that underpin a barrier to cell entry for some bat SL-CoVs and uncover genetic pathways through which this barrier might be overcome by natural selection. They also point to the existence of fitness trade-offs that might alter the evolutionary trajectory of these viruses during host-to-host transmission and identify potential Achilles' heels that could be targeted with countermeasures.

## Results

### A single point mutation in S2 affords successful rescue of rVSV bearing the SHC014-CoV bat CoV spike

We sought to generate a panel of replication-competent rVSVs bearing the spikes of bat-origin CoVs as their only entry glycoproteins. We initially selected four sarbecoviruses from clades 1, 2, and 3 with potential for zoonotic transmission in humans—WIV-1-CoV, SHC014-CoV, RmYN02-CoV, and BtKY72-CoV [4,38,39]. Previous work has shown that truncations of ≥21 amino acids in the cytoplasmic tails of divergent betacoronavirus spikes enhance the infectivities of VSV and retrovirus surrogates bearing them [40,41]. Accordingly, we replaced the VSV

A

| rVSV-CoV S | Genotype | Rescue attempts | Successful rescues |
|---|---|---|---|
| SHC014 | WT | 10 | 1* |
| SHC014 | A835D | 5 | 5 |
| RmYN02 | WT | 9 | 0 |
| RmYN02 | A806D | 1 | 1 |
| BtKY72 | WT | 9 | 0 |
| BtKY72 | A837D | 1 | 1 |
| WIV-1 | WT | 1 | 1 |
| WIV-1 | A835D | 1 | 1 |

B

C

**Fig 1. SHC014-CoV S A835D allows for rVSV rescue and is conserved amongst sarbecoviruses. (a)** Summary of recovery attempts of rVSVs bearing SHC014-CoV spike proteins. Asterisk refers to rescue that yielded the A835D substitution. **(b)** Supernatants from 293FT cells co-transfected with plasmids encoding for VSV genomes expressing eGFP and WT or A835D variants of SHC014-CoV spike as well as helper plasmids, were used to infect Vero cells. Representative images show eGFP expression in Vero cells at indicated time points (days post-infection [dpi]). Scale bar, 100 μm. **(c)** Alignment of amino acid sequences in the FPPR region (rounded rectangle) for selected coronavirus spike proteins. Subgenera of the genus *Betacoronavirus* are indicated in italics (*Sarbecovirus*, *Hibecovirus*, *Nobecovirus*, *Merbecovirus*, *and Embecovirus*). Sarbecoviruses are color-coded by clade (1a: SARS-CoV–like, red; 1b: SARS-CoV-2–like, green; 2: Southeast Asian bat-origin CoV, blue; 3: non-Asian bat-origin CoV, purple). Spikes investigated in the current study are in bold.

glycoprotein G gene with each CoV S gene engineered to encode a spike protein lacking 21 residues at its C–terminus. We also inserted a sequence encoding the enhanced green fluorescent protein (eGFP) as an independent transcriptional unit at the first position of the VSV genome, as described previously [42,43].

We attempted to generate each rVSV using a well-established plasmid-based rescue system [42,44–46]. rVSV-WIV-1-CoV S rescued readily, consistent with previous work using coronavirus reverse genetics and describing WIV-1-CoV's capacity to use ACE2 from a wide variety of hosts, including humans, civets, and Chinese horseshoe bats [4]. By contrast, numerous attempts failed to yield rVSVs for RmYN02-CoV and BtKY72-CoV (Fig 1A). rVSV-SHC014-CoV S was similarly challenging to generate; however, a single experiment

yielded a replicating viral stock. Nanopore DNA sequencing of an RT-PCR product derived from this viral population revealed that a single point mutation that results in an A835D substitution in the spike protein (corresponding to position 852 in the SARS-CoV-2 spike), had reached fixation in the population (S1 File). Sanger sequencing confirmed the presence of A835D as the only mutation in all six plaques isolated from this population. Importantly, a VSV cDNA clone bearing SHC014-CoV S(A835D) was successfully recovered in each of five attempts, whereas its WT counterpart was not (Fig 1A and 1B), providing evidence that A835D drives the generation and propagation of rVSV-SHC014-CoV S.

## Cognate substitutions facilitate the rescue of rVSVs bearing spikes from divergent bat sarbecoviruses

Amino acid sequence alignment of SHC014-CoV S with those of divergent coronavirus spikes indicated that residue 835 is in the FPPR of S2 (Fig 1C). The FPPR forms part of an intersubunit interface that is largely buried in the pre-fusion trimer, is highly conserved, and is proposed to play a key role in regulating spike conformation [47]. Most of the betacoronavirus spike sequences we examined, including all sarbecoviruses and merbecoviruses, bore an Ala at the corresponding residue (Fig 1C). By contrast, hibecoviruses, nobecoviruses, and embecoviruses do not. Accordingly, we postulated that a substitution cognate to A835D in the FPPR of other bat-origin sarbecovirus spikes could also enhance entry in the rVSV backbone. Consistent with this hypothesis, we could recover rVSV-RmYN02-CoV S(A806D) and rVSV-BtKY72--CoV S(A837D), even as their WT counterparts failed to rescue (Fig 1A). Thus, the effect of this amino acid substitution in the spike FPPR appears generalizable to multiple bat CoVs, at least in the genetic and structural context of spikes belonging to the sarbecovirus subgenus.

## A835D greatly enhances the infectivity of authentic coronavirus-like vectors

We next evaluated the possibility that A835D arose as a spike adaptation specific to the 21-amino acid cytoplasmic tail truncation and/or the heterologous VSV context. To investigate this, we used a recently developed SARS-CoV-2 replicon-based system to generate and evaluate single-cycle replicon-delivery particles (RDPs) trans-complemented with full-length WT or A835D SHC014-CoV spikes [48]. We also generated single-cycle VSV pseudotypes (scVSV) bearing WT or A835D SHC014-CoV spikes, as described previously [43,49]. We compared the infectivities of the VSV (Fig 2A) and RDP (Fig 2B) preparations in DBT-9 murine astrocytoma cells previously shown to be ACE2-null and engineered to ectopically express human and intermediate horseshoe bat (*Rhinolophus affinis*) orthologs of ACE2 (*Hs*ACE2 and *Ra*ACE2, respectively) [12,50]. Neither the VSVs nor the RDPs could transduce the parental ACE2-null DBT-9 cells, indicating that they require ACE2. This is consistent with previous findings that the SHC014-CoV spike recognizes ACE2 from humans, civets, and horseshoe bats and uses them as entry receptors [13,20]. However, ACE2 overexpression also had little effect on cell entry by both VSVs and RDPs bearing the SHC014-CoV(WT) spike. By contrast, the SHC014-CoV(A835D) spike mediated dose-dependent increases in viral entry, particularly in the cells expressing human and horseshoe bat ACE2 (Fig 2). Further, rVSV-WIV-1-CoV S(A835D) also exhibited enhanced ACE2-dependent entry relative to WT (S1 Fig). We conclude that the infection-enhancing activity of the A835D mutation is autonomous to the spike protein for multiple sarbecoviruses and can be generalized to authentic CoV virions bearing full-length spikes.

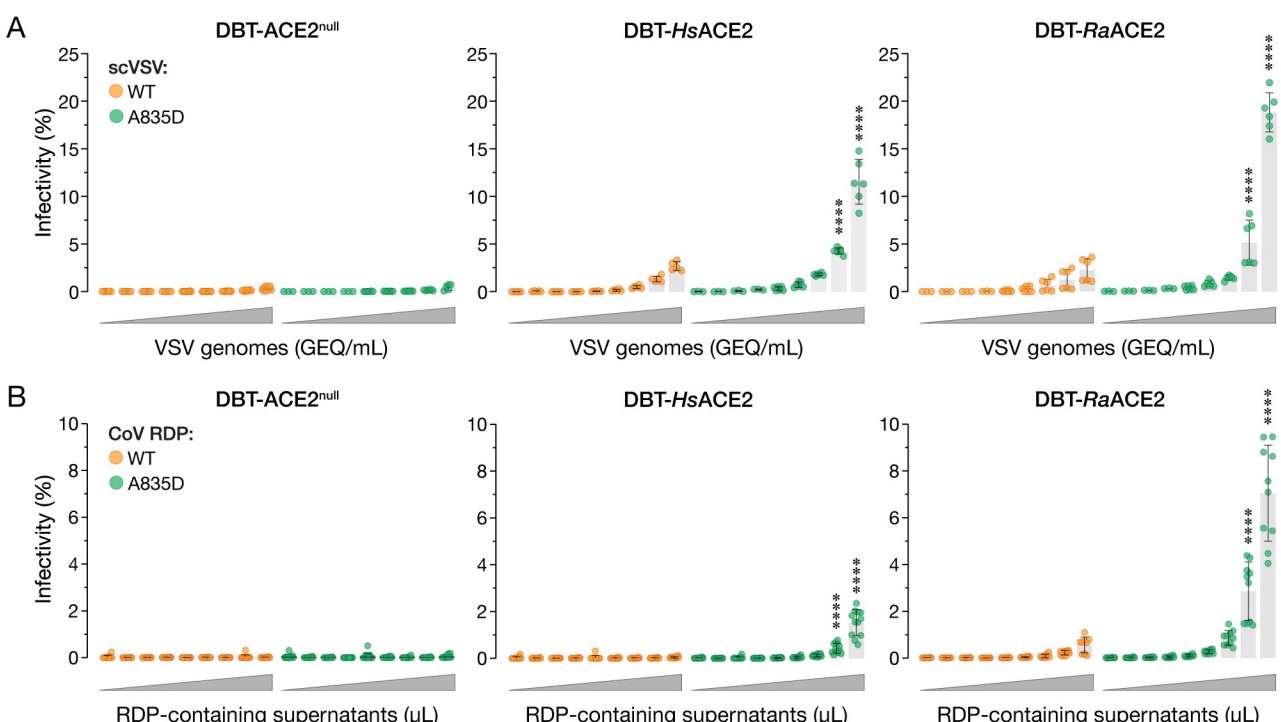

**Fig 2. A835D increases infectivity of cells in an ACE2-dependent manner. (a)** Parental DBT-9 cells or DBT-9 cells overexpressing *Hs*ACE2 or *Ra*ACE2 were infected with pre-titrated amounts of scVSV-SHC014-CoV WT or scVSV-SHC014-CoV A835D. Infection was scored by eGFP expression at 16–18 hours post-infection (average±SD, n = 5–8 from 3 independent experiments). A range of $7.2×10^2$ to $4.5×10^6$ viral genome-equivalents (GEQ) was used. **(b)** Parental DBT-9 cells or DBT-9 cells overexpressing *Hs*ACE2 or *Ra*ACE2 were infected with CoV RDPs bearing WT or A835D SHC014-CoV spikes. Infection was scored by mNeonGreen expression at 16–18 hours post-infection (average±SD, n = 10–12 from 4–5 independent experiments). Ranges of $8.2×10^3$ to $1.8×10^7$ GEQ for WT and $6.1×10^3$ to $4.0×10^7$ for A835D were used. Infectivity (%) shown is the proportion of infected cells to the total number of cells for each viral dilution. Groups (WT vs. mutant for each cell line) were compared with Welch's t-test with Holm-Šídák correction for multiple comparisons. ns p>0.05; ** p<0.01; *** p<0.001; **** p<0.0001. Only the statistically significant comparisons are shown.

## A835D does not alter spike expression or VSV incorporation

We postulated that the A835D substitution acts in part by increasing the cellular expression and viral incorporation of the SHC014-CoV spike. To test this hypothesis, we transfected 293T cells with plasmid expression vectors encoding the WT and A835D spikes. We observed similar levels of WT and A835D full-length spikes, both in total and at the cell surface (S2A Fig), indicating that A835D does not enhance SHC014-CoV spike protein expression or steady-state plasma membrane localization. Despite this, scVSVs bearing the A835D spike were much more infectious than their WT counterparts in Vero cells, recapitulating the phenotypes observed in the rVSV and CoV RDP systems (S2C Fig).

We next used an ELISA to directly assess spike incorporation into viral particles. Specifically, scVSV-coated plates were probed with antibodies specific for the SARS-CoV-2 spike and the VSV internal matrix protein, M, and the Spike-to-M ratio was used as a measure of the relative incorporation of spike into virions. A835D had little or no effect in this assay (S2B Fig). Taken together, these findings suggest that A835D exerts its effects at a post-assembly step in the viral life cycle, likely during cell entry.

## A835D enhances SHC014-CoV spike:ACE2 binding

The preceding findings in cell lines overexpressing ACE2 suggested that A835D acts at least in part by enhancing the capacity of the SHC014-CoV spike to use ACE2 for cell entry.

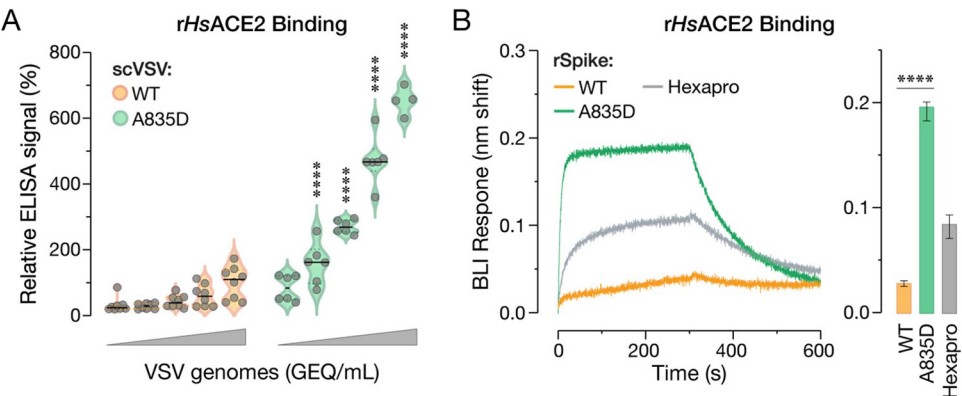

**Fig 3. A835D increases SHC014-CoV S binding to *Hs*ACE2. (a)** Genome-normalized amounts of scVSV particles bearing WT or A835D SHC014-CoV spike were diluted with serial 3-fold dilutions and loaded on an ELISA plate precoated with soluble *Hs*ACE2 and detected with a spike-specific mAb followed by an anti-human HRP-conjugated secondary antibody (average±SD, n = 6–8 from 3–4 independent experiments). A range of $1.6 \times 10^6$ to $1.3 \times 10^8$ viral GEQ was used. ELISA signal values were normalized to the highest absorbance for scVSV-SHC014-CoV WT for each replicate. Groups (WT vs. mutant) were compared with Welch's t-test with Holm-Šídák correction for multiple comparisons. ns p>0.05; ** p<0.01; *** p<0.001; **** p<0.0001. **(b)** Sensorgrams of SHC014-CoV variant spikes and a pre-fusion SARS-CoV-2 spike stabilized with 6 proline substitutions (HexaPro) binding to *Hs*ACE2 by BLI. Similar response levels of spike were captured using an anti-T4 fibritin (foldon) antibody bound to BLI biosensors and subsequently dipped into wells containing 0.5 μM *Hs*ACE2. Traces of only one replicate of three are shown for clarity. **(c)** Graph of BLI binding response collected in triplicate for the indicated SHC014-CoV spike variants (average±SD). WT and A835D were compared by one-way ANOVA, with Dunnett's correction for multiple comparisons. ns p>0.05; ** p<0.01; *** p<0.001; **** p<0.0001. Only the statistically significant comparisons are shown.

Accordingly, we considered the possibility that, despite its location at a distance from the RBD, A835D may nonetheless augment spike:ACE2 recognition. To test this hypothesis, we first evaluated the capacity of *Hs*ACE2 to capture scVSV-SHC014-CoV S particles in an ELISA. Soluble *Hs*ACE2-coated plates were incubated with genome-normalized amounts of scVSVs, and particle capture was detected with a CoV spike-specific monoclonal antibody (mAb). Despite previous evidence that the isolated SHC014-CoV spike RBD recognizes *Hs*ACE2 with high avidity [20], WT SHC014-CoV spike could capture *Hs*ACE2 only poorly, whereas its A835D counterpart exhibited a dramatic, dose-dependent increase in *Hs*ACE2 capture (Fig 3A).

We next deployed an established biolayer interferometry (BLI) assay to measure binding of the WT and A835D spikes to *Hs*ACE2. Because the A835D substitution is not located in the receptor-binding motif, the binding response can be used as a proxy for RBD exposure or 'availability' to bind ACE2, as shown previously [51]. We began by producing trimeric ectodomains of both spike proteins with a C–terminal T4 fibritin (foldon) domain as described previously for other spikes [52]. Then, recombinant SHC014-CoV spike ectodomains were captured to similar levels onto BLI probes using an anti-foldon IgG and subsequently dipped into solutions containing *Hs*ACE2. A SARS-CoV-2 spike stabilized with six prolines (Hexapro) served as a positive control and efficiently captured *Hs*ACE2 from solution, as expected [53]. In contrast, WT SHC014-CoV spike bound weakly to *Hs*ACE2, and its activity was substantially boosted by the A835D substitution, corroborating our ELISA results (Fig 3B and 3C).

Finally, to further investigate the effect of A835D on the conformation of the SHC014-CoV spike, we assessed the binding of a panel of spike-specific monoclonal antibodies (mAbs) by scVSV-SHC014-CoV S ELISA, as above (S3 Fig). Adagio-2 (ADG-2), an RBD binder that can only recognize the spike in its open, "RBD-up" conformation [54], resembled HsACE2 in

showing greatly enhanced binding to the A835D SHC014-CoV spike compared to WT. By contrast, no such difference was observed with S309, an RBD-specific mAb that can recognize both open and closed spikes [55,56], and with two S2-directed antibodies (S3 Fig). Together, these findings strongly suggest that the A835D substitution augments the SHC014-CoV spike's capacity to bind *Hs*ACE2 by increasing the availability of its RBDs for receptor engagement, but without causing large-scale changes in the pre-fusion conformation of the spike.

## High-resolution structure of a SHC014-CoV spike ectodomain

We performed cryo-electron microscopy (cryo-EM) studies to define the interactions made by residue 835 in the WT SHC014-CoV spike. Purified, soluble spike ectodomain was applied to a cryo-EM grid and vitrified. Sufficient particle density with little aggregation was observed, and a dataset containing 2,025 micrographs was collected (S1 Table). Extracted particles were subjected to two-dimensional (2D) classification, which resulted in class averages populated with fully closed trimers, wherein all three RBDs are in the 'down' position. This finding stands in contrast to previous cryo-EM studies of SARS-CoV-2 spike ectodomains, in which substantial proportions of partially open spikes with at least one RBD in the 'up' position were observed [57–59]. A 3.1-Å–resolution 3D reconstruction of the WT SHC014-CoV spike trimer was obtained and found to resemble that of the closed SARS-CoV-2 spike (Figs 4A, S4 and S5), which was expected given the ≈80% sequence identity between the proteins. The flexibility of the NTD led to poorly resolved regions of the map for this domain, preventing the modeling of some loops based on our reconstruction. Additionally, the FP was not visible in our cryo-EM map, as also observed in many SARS-CoV-2 spike structures. In all, we were able to build residues 33–1124, excluding 45–46, 71–78, 94–97, 121–123, 134–155, 167–182, 205–212, 237–251, 268–270, 617–621, 665–671, and 812–831. C–terminal FPPR residues were resolved (residues 832–840), affording an examination of residue A835 and its local environment.

A835 is part of an α-helix proximal to the FP, and it lies at the interface between protomers (buried surface area upon oligomerization: 22.7%). The A835 sidechain is pointed inward toward V946 in the S2 heptad repeat 1 (HR1) of the same protomer and toward residues D555, V556 and S557 in the S1 subdomain I (SD1) of the neighboring protomer (Fig 4B). Residues 555–557 in SD1 connect A835 to the RBD in the neighboring protomer (Fig 4D). Structure-based *in silico* analyses consistently suggested that A835D substitution increases the binding free energy between protomers, substantially affecting the stability of the trimer ($\Delta\Delta G_{trimer}$: > 3 Kcal/mol; $\Delta\Delta G_{interface}$: +1.25 Kcal/mol by FoldX; +1.08 Kcal/mol by flexddG). We propose that the introduction of a negatively charged residue at position 835 decreases the stability of the spike trimer by disrupting a network of hydrophobic interactions within (A835–V963) and between protomers (A835–V556; Fig 4B) and triggering unfavorable electrostatic interactions between protomers by virtue of its close proximity to an electronegative surface region in the neighboring protomer (Fig 4C). In support of this hypothesis, scVSVs bearing a Glu at position 835 (A835E) showed a similar enhancement of infectivity over WT as A835D (S6 Fig). Further evidence for the reduced stability of the A835D spike includes a decrease in the expression of its ectodomain relative to WT (S7A Fig). Moreover, the A835D spike ectodomain was found to be unstable on cryo-EM grids with mostly dissociated particles and few trimeric spikes, thereby preventing high-resolution structure determination (S7B and S7C Fig). Together, these results strongly suggest that the A835D substitution has a destabilizing effect on the pre-fusion conformation of the SHC014-CoV spike, resulting in increased transition of RBDs from 'down' to 'up' positions and enhanced ACE2 binding.

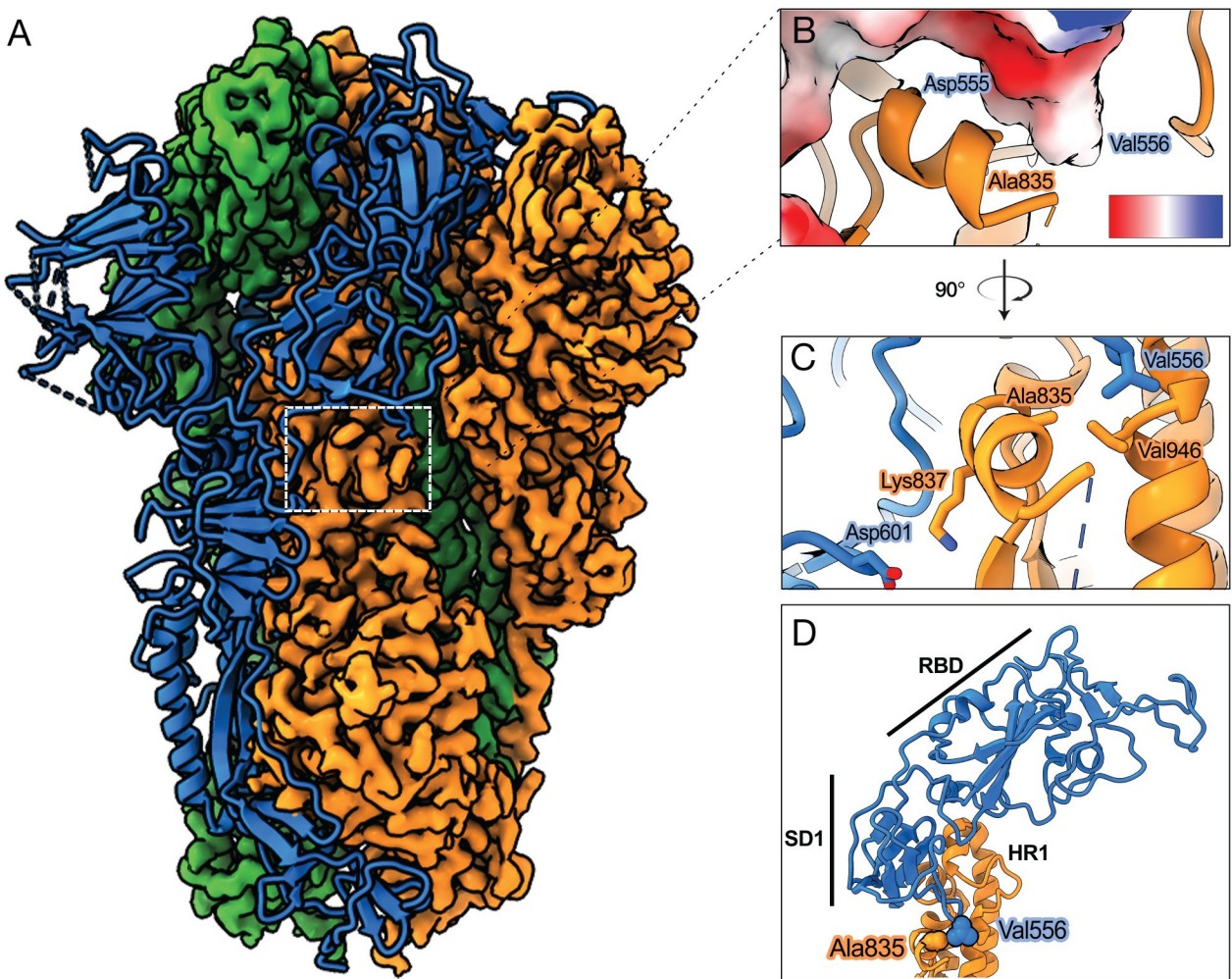

**Fig 4. Structural determination of SHC014-CoV spike. (a)** Cryo-EM map and model of the WT SHC014-CoV spike. The cryo-EM map is shown for two protomers and the third protomer is shown as a ribbon representation. **(b)** Ala835 is proximal to Asp555 of the neighboring protomer, which is colored by electrostatic surface potential, red (negative) to blue (positive). Scale bar, -10 to +10 kT/e. **(c)** Ala835 makes van der Waals interactions with Val556 of a neighboring protomer (blue) and Val946 of the same protomer (orange). Lys837 (orange) makes a salt bridge with Asp601 of a neighboring protomer (blue). **(d)** S2 of one protomer (orange) and subdomain 1 (SD1) and RBD of an adjacent protomer (both in blue) are shown as ribbons to depict the distance between the Ala835-containing hydrophobic pocket to the RBD of the adjacent protomer. Labelled residues are shown as spheres.

## A835D reduces spike thermostability and alters the temperature dependence of RBD availability

To test the hypothesis that A835D destabilizes the SHC014-CoV spike, we measured the temperature dependence of viral infectivity loss (inactivation) as a surrogate for the thermal stability of the viral membrane-embedded pre-fusion spike, as we described previously for VSVs bearing filovirus glycoproteins [60]. scVSV particles bearing WT and A835D SHC014-CoV spikes were preincubated at different temperatures, then shifted to 4°C, and titrated for infectivity on DBT-*Ra*ACE2 cells. WT spike-containing particles suffered loss of infectivity between 48.5–51°C, with no detectable infectivity at 53°C. We observed a left shift in temperature-dependent infectivity loss for A835D—the mutant particles underwent inactivation between 41–46°C, with no detectable infectivity at 48.5°C (Fig 5A). Thus, the A835D substitution

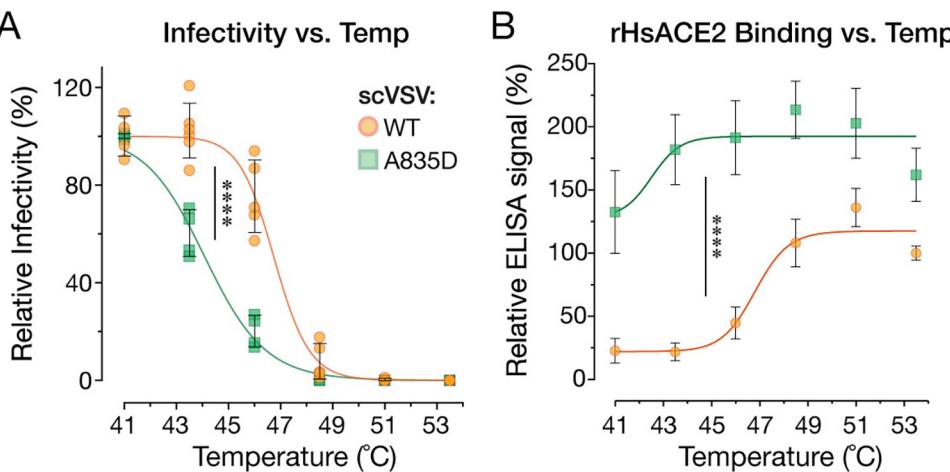

**Fig 5. A835D reduces thermostability of the SHC014-CoV spike. (a)** scVSV-SHC014-CoV S particles were incubated at various temperatures for 1 h and used to infect DBT-9 cells overexpressing *Ra*ACE2. (average±SD, n = 4–6 from 2–3 independent experiments). Infectivity levels were normalized to the infectivity percentage at 41˚C for each virus. Area under the curve (AUC) values were calculated for each curve, and groups were compared by one-way ANOVA with Dunnett's correction for multiple comparisons. **(b)** Pre-titrated amounts of scVSV-SHC014-CoV S particles were incubated at various temperatures for 1h and loaded onto *Hs*ACE2-coated ELISA plates. A spike-specific mAb and anti-human HRP-conjugated secondary antibody were used to detect the spike protein. $9.2\times10^6$ viral GEQs per well were used. (average±SD, n = 12 from 6 independent experiments). ELISA signals were normalized to the absorbance observed for scVSV-SHC014-CoV WT at 53.5˚C. Groups were compared by two-way ANOVA with Tukey's correction for multiple comparisons, ns p>0.05; ** p<0.01; *** p<0.001; **** p<0.0001.

renders the SHC014-CoV spike more susceptible to thermal inactivation in the context of intact viral particles.

In parallel experiments, we measured the activity of scVSVs bearing WT and A835D SHC014-CoV spikes in the *Hs*ACE2 capture ELISA as a function of temperature. We observed an increase in spike:ACE2 binding for WT particles between 43.5–48˚C, a temperature range just below and overlapping that of viral inactivation for infection. Strikingly, A835D particles exhibited higher levels of *Hs*ACE2 binding at all temperatures tested, with an increase that titrated between 41–43.5˚C, consistent with their temperature range for inactivation (Fig 5B). We conclude that the A835D substitution has a destabilizing effect on the pre-fusion conformation of the SHC014-CoV spike, sensitizing it to undergo conformational changes in response to heat that result in enhanced RBD exposure and culminate in viral inactivation. We surmise that this phenotype, together with data in Figs 4 and 5, reflects the increased propensity of the A835D spike to sample open conformers with RBDs in the 'up' position at physiological temperatures (see below).

## NTD substitution F294L enhances viral entry and ACE2 recognition by WT and A835D spikes

In seeking to identify key intra-spike interactions altered by the A835D substitution while structural studies on the SHC014-CoV spike (Fig 4) were in progress, we modeled both WT and A835D SHC014-CoV spikes using AlphaFold2 [61,62]. These studies suggested that A835D may engage in a new polar interaction with residue T842 in the FPPR. Although we could not structurally corroborate this hypothesis given the relative instability of the A835D ectodomain, we nonetheless attempted to generate an rVSV bearing an SHC014-CoV(A835D/T842A) double-mutant spike. As with the WT spike, the double mutant was challenging to rescue, but one attempt yielded a replicating viral population. Nanopore sequencing of viral

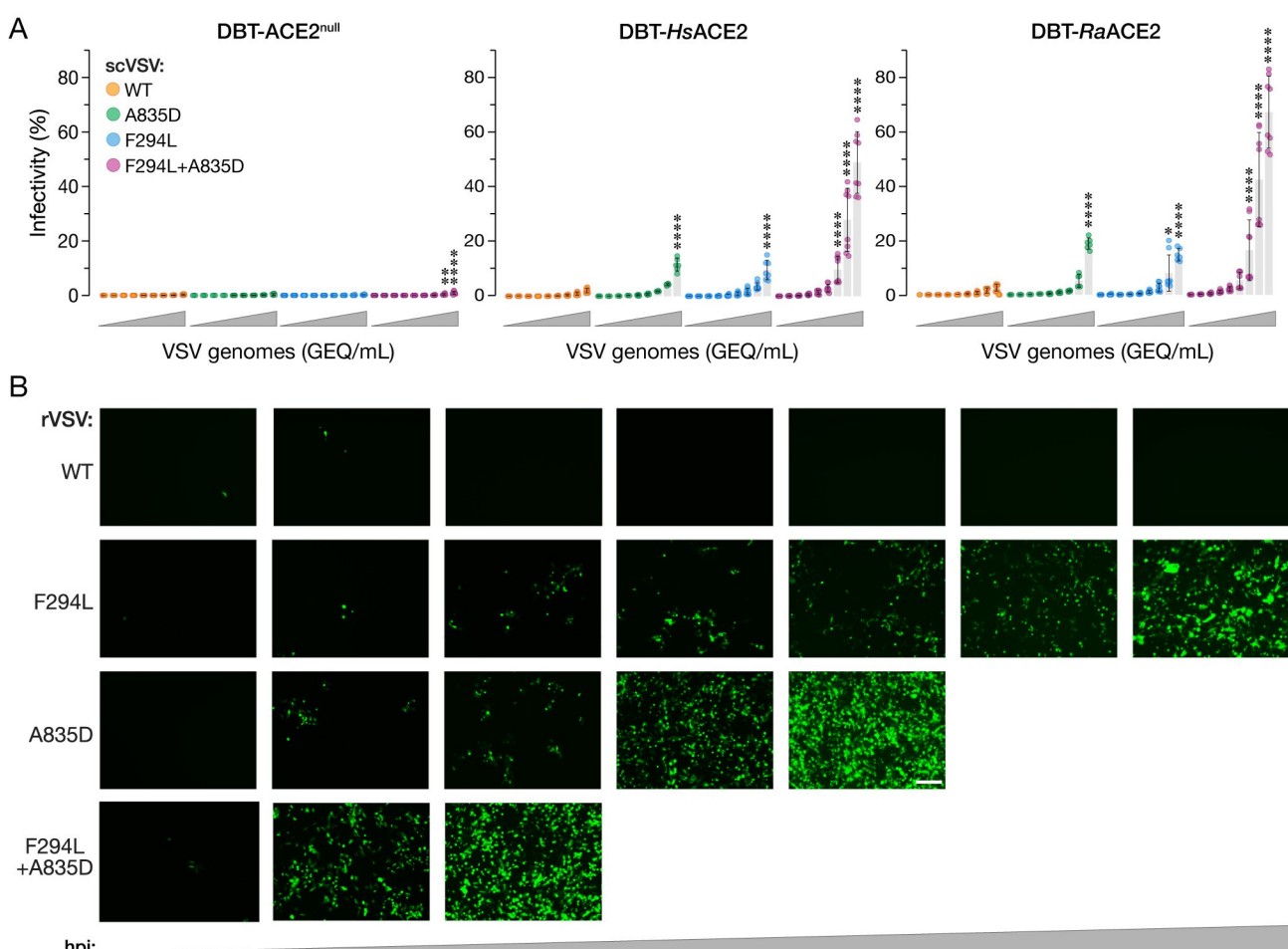

**Fig 6. A835D and F294L substitutions enhance infectivity. (a)** Parental DBT-9 cells or DBT-9 cells overexpressing *Hs*ACE2 or *Ra*ACE2 were infected with pre-titrated amounts of scVSV-SHC014-CoV particles bearing WT, A835D, F294L, or F294L+A835D spike. Infection was scored by eGFP expression at 16–18 hours post-infection (average±SD, n = 5–8 from 3 independent experiments). A range of $1.5 \times 10^3$ to $9.6 \times 10^6$ viral GEQ was used. Infectivity (%) shown is the proportion of infected cells to the total number of cells for each viral dilution. Groups (WT vs. mutant for each cell line) were compared with Welch's t-test with Holm-Šídák correction for multiple comparisons. ns p>0.05; ** p<0.01; *** p<0.001; **** p<0.0001. Only the statistically significant comparisons are shown. **(b)** Supernatants from 293FT cells co-transfected with plasmids encoding for VSV genomes expressing eGFP and WT, A835D, F294L, or F294L+A835D variants of SHC014-CoV spike as well as helper plasmids, were used to infect Vero cells. Representative images show eGFP expression in Vero cells at 24, 60, 108, 204, 252, 276, and 273 h post-infection [hpi]. Representative images are shown. Scale bar, 100 μm.

cDNA from this population indicated that both parental mutations were present but uncovered one additional point mutation in the spike NTD in most reads (S2 File), causing the F294L substitution (corresponding to position F306 in the SARS-CoV-2 spike). Sanger sequencing confirmed the acquisition of F294L and retention of both parental substitutions in all three plaques isolated from this population, strongly suggesting that F294L and/or T842A played functional roles in rVSV recovery.

To dissect the effects of these substitutions on viral entry, we generated scVSVs bearing WT and mutant SHC014-CoV spikes. Spikes containing F294L alone exhibited increases relative to WT in cell entry (Fig 6A) and *Hs*ACE2 binding (Fig 7A) resembling those observed for A835D. F294L+A835D conferred a further enhancement in infectivity relative to the single mutants in DBT-9 cells over-expressing *Hs*ACE2 and *Ra*ACE2 (Fig 6A). Because F294L was originally selected in the A835D/T842A background, we additionally tested the infectivity of

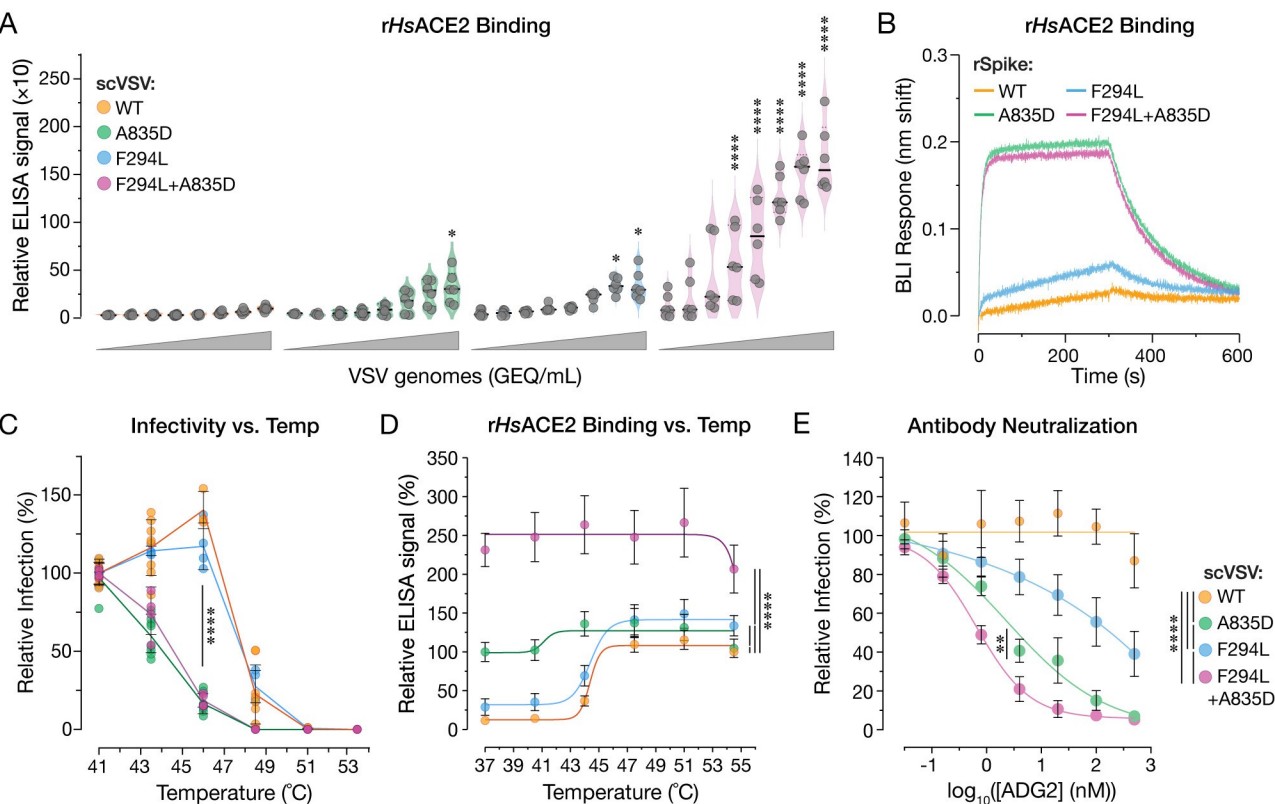

**Fig 7. A835D and F294L enhance ACE2 binding but act through distinct mechanisms. (a)** Genome normalized amounts of scVSV particles bearing WT, A835D, F294L, or F294L+A835D SHC014-CoV spike were diluted with serial 3-fold dilutions and loaded on an ELISA plate precoated with soluble *Hs*ACE2 and detected with a spike-specific mAb followed by an anti-human HRP-conjugated secondary antibody (average±SD, n = 6–8 from 3–4 independent experiments). A range of $5.5×10^3$ to $1.2×10^7$ viral GEQ was used. ELISA signals were normalized to the absorbance observed for scVSV-SHC014-CoV WT at the highest number of viral genomes. Groups (WT vs. mutant) were compared with Welch's t-test with Holm-Šídák correction for multiple comparisons. ns p>0.05; ** p<0.01; *** p<0.001; **** p<0.0001. **(b)** Sensorgrams of SHC014-CoV spike variants binding to *Hs*ACE2 by BLI. Spike variants were captured to similar response levels using an anti-T4 fibritin antibody bound to BLI biosensors and subsequently dipped into wells containing 0.5 µM *Hs*ACE2. One of three replicates is shown for clarity. **(c)** scVSV-SHC014-CoV S particles were incubated at various temperatures for 1h and used to infect DBT-9 cells overexpressing *Ra*ACE2. Infection levels were normalized to values observed at 41˚C for each virus (average±95%CI, n = 4–8 from 2–4 independent experiments). AUC values were calculated for each curve, and groups were compared by one-way ANOVA with Dunnett's correction for multiple comparisons. Statistical significance is shown for WT vs. A835D, WT vs. F294L+A835D, F294L vs. A835D, and F294L vs. F294L+A835D. Differences among all other two-way comparisons were not statistically significant **(d)** Pre-titrated amounts of scVSV-SHC014-CoV S particles were incubated at various temperatures for 1 h and loaded onto *Hs*ACE2-coated ELISA plates. A spike-specific mAb and anti-human HRP-conjugated secondary antibody were used to detect the spike protein. Absorbance values were normalized to that observed for scVSV-SHC014-CoV WT particles at 54.5˚C. (average±95%CI, n = 12 from 3 independent experiments). $5.1×10^6$ viral GEQs were used per well. AUC values were calculated for each curve, and groups were compared by one-way ANOVA with Dunnett's correction for multiple comparisons. **(e)** Pre-titrated amounts of scVSV-SHC014-CoV S bearing WT, A835D, F294L, or F294L+A835D spikes were incubated with serial 3-fold dilutions of mAb ADG-2 starting at 500 nM, at 37˚C for 1 h. Virus:mAb mixtures were applied to monolayers of DBT-9 cells overexpressing *Ra*ACE2 cells. At 18 h post-infection, cells were fixed, nuclei were counterstained, and infected cells were scored by eGFP expression (average±SD, n = 9 from 3 independent experiments). Infectivity levels were normalized to the infectivity percentage with no mAb present for each virus. AUC values were calculated for each curve, and groups were compared by one-way ANOVA with Dunnett's post hoc test, ns p>0.05; ** p<0.01; *** p<0.001; **** p<0.0001. Only the statistically significant comparisons are shown.

scVSVs bearing SHC014-CoV spikes with A835D+T842A and F294L+A835D+T842A (S8 Fig). We saw no significant difference in infectivity when T842A was introduced into scVSV-SHC014-CoV S particles bearing A835D or F294L+A835D. These findings strongly suggested that F294L, but not T842A, afforded a growth advantage to viral clones in the rVSV rescue experiments.

Accordingly, we attempted to rescue rVSVs with SHC014-CoV(F294L) and SHC014-CoV (F294L+A835D) spikes in parallel with rVSV-SHC014-CoV(A835D) and WT spikes. Rescues

of rVSV-SHC014-CoV bearing spikes with F294L, A835D, and F294L+A835D were all successful, with F294L+A835D rescuing the fastest, followed by A835D, and then F294L (Fig 6B). As we observed previously (Fig 1B), WT did not rescue. Sanger sequencing of the spike gene in isolated plaques confirmed the presence of only these mutations in each respective virus prep.

The boosts in infectivity and rVSV rescue seen in the context of F294L+A835D were also associated with a commensurate, dramatic gain in *Hs*ACE2 binding to viral particles at 37˚C (Fig 7A). BLI analyses of *Hs*ACE2 binding to SHC014-CoV spike ectodomains bearing F294L and F294L+A835D yielded largely concordant results, but with some differences (Fig 7B). Specifically, although F294L did significantly enhance spike:*Hs*ACE2 binding, its effect was smaller than that observed for A835D. Further, F294L+A835D did not boost *Hs*ACE2 binding over A835D alone in the BLI assay. These discordances between the virus- and spike ectodomain-based ACE2 binding assays may simply reflect the greater instability of the ectodomains bearing A835D relative to viral membrane-embedded spikes—juxtamembrane and transmembrane sequences in the latter are predicted to counteract the destabilizing effects of the ectodomain substitutions [63–65]. Alternatively, or in addition, they may reflect total saturation of the BLI biosensors, preventing further increases in RBD availability from being measured.

Finally, we considered the possibility that F294L+A835D constitutes an intramolecular epistasis, in which the combination of the two substitutions affords significantly different fitness than the sum of their effects when considered independently. Specifically, we used the scVSV infectivity (DBT-*Hs*ACE2; Fig 6A) and *Hs*ACE2 binding (Fig 7A) datasets as proxies for viral fitness to assess the likelihood of an epistatic interaction between F294L and A835D with the linear-modeling based approach of Miton and colleagues [66]. We found that our data fit best to a first-order model in which the enhanced biological activity of the double-mutant relative to its single-mutant parents could be largely explained as the additive effect of the independent contributions of each substitution (S2 Table). We conclude that F294L and A835D independently enhance ACE2 recognition and spike-dependent entry, with their combined effect providing the double mutant a selective growth advantage over the single mutants (Fig 6B).

## F294L and A835D act through distinct mechanisms

The location of F294L in the NTD, at a substantial distance from the receptor-binding motif, indicated that its effect on spike:*Hs*ACE2 binding is indirect, and as with A835D, likely mediated by a change in the RBD availability of pre-fusion spikes. To begin to uncover F294L's mechanism of action, we investigated its effects on spike thermal stability in both the WT and A835D contexts. As described above (Fig 5A), scVSV particles bearing each spike were preincubated at different temperatures and then titrated for infectivity on DBT-*Ra*ACE2 cells (Fig 7C). Unlike with A835D, we observed little or no left shift in the temperature dependence of thermal inactivation for F294L alone, relative to WT. Moreover, the F294L+A835D double mutant resembled its A835D parent in thermal stability, sharing a large left shift in thermal inactivation, as observed in Figs 5A and 7C. Therefore, F294L does not contribute to spike destabilization, which is concordant with our observation that the F294L spike ectodomain resembles its WT counterpart in expression level and pre-fusion stability. Expression levels were compared by overlaying absorbance traces from size-exclusion chromatography (SEC) of different variants. All cultures were prepared in parallel in a similar manner, allowing for direct comparison (S7A Fig).

Finally, we examined the capacity of F294L to alter RBD availability in a temperature-dependent manner by *Hs*ACE2 capture ELISA (Fig 7D). Consistent with its behavior in the

thermal inactivation studies (Fig 7C), F294L afforded an increase in $Hs$ACE2 binding that titrated over a similar temperature range as WT. Strikingly, however, the F294L+A835D double mutant exhibited a large increase in $Hs$ACE2 binding at all temperatures tested, with no discernible temperature-dependent effect between 37–55˚C. We conclude that F294L enhances RBD availability via a mechanism distinct from that of A835D. Moreover, the additive effects of F294L and A835D afford the double mutant a substantial increase in RBD availability at physiological temperature and an attendant enhancement in viral infectivity relative to WT and its single-mutant counterparts.

### RBD-mobilizing substitutions enhance viral sensitivity to antibody neutralization

Previous work with SARS-CoV-2 variants of concern has shown that amino acid substitutions which modulate spike dynamics can alter the exposure of neutralizing antibody epitopes [67–69]. The large increases in RBD availability mediated by A835D and F294L suggested that these substitutions might also impact viral susceptibility to antibody neutralization. To investigate this possibility, we measured the activity of ADG-2, a broad neutralizer of clade 1 sarbecoviruses, which binds an RBD epitope accessible only in the 'up' RBD [54,70] against VSVs bearing WT and mutant SHC014-CoV spikes. We observed little neutralization of WT at mAb concentrations as high as 500 nM. In contrast, A835D, F294L, and A835D+F294L spikes were all neutralized by ADG-2 with sensitivities proportional to their capacity to recognize ACE2 (Fig 7A) and mediate entry (A835D+F294L > A835D > F294L) (Fig 6B). Remarkably, incorporation of both substitutions into the SHC014-CoV spike afforded an enhancement in neutralization $IC_{50}$ of >500-fold (Fig 7E). These findings provide additional evidence that A835D and F294L enhance viral entry by increasing RBD exposure. Moreover, they suggest that substitutions of this type would be subject to a substantial fitness tradeoff in the form of enhanced sensitivity to neutralizing antibodies, with likely implications for their evolutionary trajectories during adaptation to new hosts following zoonotic transmission.

## Discussion

Bat-origin CoVs with high sequence similarity to their human outbreak-causing counterparts pose continuing risks to human populations. Consequently, considerable effort has been expended in understanding their biology and uncovering the molecular determinants of their spillover potential. Because few authentic CoVs isolated from nature or recovered from molecular clones are available, and generating and working with these agents requires a high biosafety setting, researchers have largely turned to surrogate systems, including single-cycle pseudotyped viruses and recombinant proteins, to study 'pre-emergent' CoV spike proteins. Herein, we systematically exploited the forward-genetic capabilities of rVSVs to investigate molecular adaptation in a panel of these spikes. We found that the successful recovery and high replicative fitness of rVSVs bearing WT spikes from SHC014-CoV was associated with the acquisition of two growth-adaptive amino acid substitutions outside the RBD. Through studies with both viral particles and recombinant spike proteins, we show that these substitutions enhance viral receptor binding and entry by regulating spike conformation, even though they are spatially separated in the three-dimensional structure of the SHC014-CoV spike (also determined herein). Our findings confirm the importance of interactions among the S1 NTD and RBD and the S2 FPPR sequences in the CoV spike in influencing virus-receptor interactions and entry [71] but also highlight the existence of distinct, context-dependent, and at least partially transferable genetic pathways for viral adaptation in the non-RBD spike sequences of bat-origin CoVs. Similar pathways may be accessible to bat-origin CoVs during spillover and

sustained host-to-host transmission. Further, they may provide diagnostic markers useful for evaluating the potential for emergence, expansion, and transmissibility of zoonotic strains associated with sporadic human cases.

Recent work suggests that multiple pre-fusion spikes from CoVs of rhinolophid bat and pangolin origin (e.g., BANAL-20-52, BANAL-20-236, RaTG13, Pangolin-CoV), have evolved to favor a more compact 'closed' state, wherein their RBDs largely sample the 'down' conformation that is incompetent for receptor binding [17,72,73]. Nonetheless, spikes derived from these bat CoVs could associate with ACE2 and mediate entry into cells expressing human and rhinolophid ACE2 orthologs, indicating that their RBDs can infrequently transit to the 'up' conformation and engage ACE2. Our structural studies on an ectodomain of the WT SHC014-CoV spike revealed only closed trimers with all three RBDs in the down position, as also observed by another recent study [74] (also see below). We should note, however, that a larger dataset with substantially more particles may allow for the identification of a small subset of spikes with RBDs in the 'up' position. In contrast to previous findings with other bat CoVs, we found that the SHC014-CoV spike bound weakly with *Hs*ACE2 and was ineffective at viral entry into cells expressing human or rhinolophid ACE2 in both VSV and CoV contexts. Together with previous observations that the isolated SHC014-CoV spike RBD binds to diverse ACE2 orthologs with high affinity [20], these findings suggested that the availability of receptor-accessible 'up' RBDs is even lower in SHC014-CoV (and likely, also in RmYN02-CoV and BtkY72-CoV, the other bat-origin CoVs examined herein). Although somewhat surprising, these results are consistent with previous observations by Menachery and colleagues that lentiviral SHC014-CoV pseudotypes have little activity in human ACE2-expressing cells [13]. Moreover, in at least one instance in which SHC014-CoV S pseudotypes were employed as part of larger CoV spike panels, they were found to have ~10-fold lower titers in *Hs*ACE2-o-verexpressing cells relative to their WIV-1–CoV counterparts; A. Balazs, personal communication). Finally, while this manuscript was under review, a study by Qiao and colleagues corroborated our observation that WT SHC014-CoV S pseudotypes are indeed poorly infectious [74]. The totality of the evidence thus supports the conclusion that, although the pre-fusion spikes of many bat-origin CoVs tend to favor the closed (all RBDs down) state, they exist along a continuum with regard to their propensity to sample conformationally open states that are competent to bind receptors.

We identified a key genetic adaptation for successful rVSV-SHC014-CoV S rescue and multiplication—the substitution A835D in the S2 FPPR. In accordance with our structural and biochemical findings, A835D boosted cell entry through substantial enhancements in human and rhinolophid ACE2 binding in both viral particle- and ectodomain-based assays. Studies on SARS-CoV-2 have highlighted the influence of a disulfide-bonded helix-turn-helix–forming sequence in the FPPR on spike conformational dynamics and RBD positioning [47,75]. Specifically, structures of full-length spikes suggested that, in its ordered state, the FPPR clamps the RBD in its 'down' position and stabilizes the closed conformation of the spike trimer. A structural transition in the FPPR from ordered to disordered, possibly favored at endosomal acidic pH, was proposed to initiate a conformational relay that repositions the RBD to the 'up' state and drives spike opening [71]. A C–terminal segment of the FPPR was ordered in our high-resolution cryo-EM structure of the WT SHC014-CoV spike, and we anticipate that A835D would disrupt a network of intra- and inter-protomer hydrophobic interactions with residues in the S1 CTD and S2 heptad repeat 1 (HR1) and trigger unfavorable electrostatic interactions between protomers, thereby affording greater movement of the FPPR and leading to increased RBD availability.

Through rVSV rescue experiments in an A835D mutant background, we identified a second entry-enhancing substitution, F294L, in the NTD of the SHC014-CoV spike. Although F294L, like A835D, could act alone by increasing RBD availability, combining the two

substitutions afforded substantial increases in both viral infectivity and RBD availability. Strikingly, F294L did not reduce spike thermostability, which we conjecture may be incompatible with assembly of the pre-fusion spike in the context of the already destabilizing A835D. In the WT spike, F294 is located near the C–terminus of the NTD (S9B Fig), and is coordinated by contacts with K266 and K288 in the adjacent NTD-RBD linker to form a 'double-cation-pi' sandwich (S9A Fig). We propose that the loss of this interaction network in the F294L spike enhances the flexibility of the NTD-RBD linker—implicated in S1 inter-domain movements in the SARS-CoV-2 spike—thereby increasing RBD availability [76].

We found that engineering of substitutions cognate to A835D in the bat-origin CoV spikes of RmYN02-CoV (clade 2) and BtKY72-CoV (clade 3) afforded the rescue of rVSVs bearing these spikes, suggesting that the spike-opening effect of this substitution is transferable to divergent sarbecovirus genetic backgrounds. Since BtKY72-CoV requires ACE2 [20,77] whereas RmYN02-CoV does not [38], this effect appears independent of receptor usage. A835D also enhanced the efficiency of the highly entry-competent WIV-1-CoV spike (S1 Fig), implying that it can function in this distinct genetic background, which affords a relatively open pre-fusion spike despite its A835/F294 genotype (also see below). Interestingly, although A835 is highly conserved among sarbecovirus spikes, the residue cognate to 294 is already polymorphic (F/L) (S9C Fig), raising the possibility that the acquisition of new substitutions in different 294 backgrounds could promote shifts in viral fitness. A more comprehensive mutational approach, of the type recently described by Dadonaite and colleagues [78–80] will likely be needed to uncover such potential epistatic effects.

Observations that the pre-fusion spikes of many bat-origin CoVs [16], as well as the four human endemic CoVs [52,81–84], predominantly assume a closed conformation in structural studies may reflect a requirement for viral replication in immune-competent hosts with pre-existing antiviral immunity. We speculate that, in bat-origin CoVs with especially stable spikes (such as SHC014-CoV), this phenomenon may also be coupled with viral adaptation to a fecal-oral transmission lifestyle in the insectivorous bat host. To wit, Ou and co-workers recently found that an entry-enhancing substitution in Laotian sarbecoviruses proposed to increase spike opening rendered viral particles more susceptible to proteolytic inactivation [17]. These authors postulated that repeated viral passage through the highly proteolytic environment of the gut may have selected for spike stability at the expense of entry efficiency. Conversely, a body of work has shown that exogenous treatment with the gut serine protease trypsin can activate many CoV spikes with monobasic S1-S2 and S2′ cleavage sites, a process that likely resembles spike activation in the gut of the natural hosts [5,19,85–87]. Given the known interplay between spike cleavage, conformational dynamics, and fusion triggering [36,72,76,88], it is thus conceivable that the extreme stability of the SHC014-CoV spike reflects a setpoint that is tuned for its activation to bind its receptor and undergo fusion triggering in the protease-rich milieu of the gut lumen. This setpoint would also be expected to reduce viral exposure to immune surveillance (see below).

Current evidence raises the possibility that such spike optimizations for replication in the natural host impose a barrier to viral transmission via the respiratory route [19,86,89,90], at least in part because the virus must adapt to use the host proteases enriched in the respiratory tract, such as the plasma membrane serine protease TMPRSS2 [91,92], instead of luminal gut proteases. Indeed, recent work indicates that clade 2 sarbecoviruses, in particular, require treatment with exogenous trypsin to infect cells [5,19,85,93]. We speculate that this behavior at least partly reflects a high degree of spike stability, similar to our findings herein with SHC014-CoV. A distinct strategy exemplified by divergent coronaviruses, most notably SARS-CoV-2 [94] and MERS-CoV [95], is the acquisition of a furin cleavage site at the S1-S2 boundary, which not only affords spike cleavage in virus-producer cells [96,97], but also appears to enhance TMPRSS2-mediated SARS-CoV-2 spike cleavage in target cells [98]. The

SARS-CoV-2 spike furin site promotes viral entry and respiratory transmission in a ferret model [99] but also destabilizes its pre-fusion conformation, likely causing the selection of D614G as a stabilizing compensatory substitution [36,100]. We posit that the novel non-RBD S1 and S2 substitutions uncovered in the current study represents yet another conceivable path for spike adaptation to a proteolytically poor environment—one that does not require spike pre-cleavage in producer cells. Finally, we cannot currently exclude the possibility that A835D+F294L also directly enhances spike cleavage(s) and/or downstream steps, such as S1 shedding, in the conformational cascade leading to membrane fusion. Substitutions in the spikes of SARS-CoV-2 and other bat-origin CoVs that contribute to spike opening (e.g., T372A in the RBD) [16,17,29] at the expense of stability may also act in a manner similar to A835D+F294L to drive viral entry. A recent study using chimeric SHC014-CoV and WIV-1-CoV spike-pseudotyped viruses identified another S1 substitution, Y623H—in the same region as D614G in the SARS-CoV-2 spike, that enhanced the infectivity of SHC014-CoV pseudoviruses [74]. Although the authors did not directly examine the ACE2-binding properties of this mutant, they provided structural evidence that it affords spike opening. Interestingly, we did not observe Y623H in our VSV system. More comprehensive approaches to examine the genetic landscape of SARS-CoV-2 spike adaptation, such as deep mutational scanning methods, have identified additional non-RBD substitutions that alter ACE2 binding by modulating RBD conformation [78,79,80]. Together, these findings reinforce the idea that multiple evolutionary pathways exist by which bat CoVs can increase (or decrease) the propensity of their RBDs to transition to the 'up' conformation.

A limitation of the experimental evolution 'sandbox' we have explored herein is its relative simplicity—viral clones in our cell culture system would have encountered only a small subset of the competitive pressures faced by their counterparts in the wild. Crucial among these countervailing pressures are those imposed by the host immune system. As a case in point, substitutions in bat-origin CoV spikes that engender a more open spike architecture might be more vulnerable to antibody neutralization due to enhanced epitope exposure, as shown for the T372A RBD mutant in SARS-CoV-2 [17]. Introduction of the T372A substitution into bat CoVs, BANAL 20–52 and BANAL 20–236, was shown to increase antibody neutralization and sensitivity to protease digestion [17]. Notably, T372 is also a highly conserved residue amongst bat CoVs, highlighting its potentially key role in the evolutionary adaptation of SARS-CoV-2 in humans. Strikingly, the omicron variant of SARS-CoV-2 appears to have evolved to favor a more closed spike, thereby reversing some of the spike-opening adaptations in earlier viral lineages, presumably in the face of antibody pressure [101,102]. In line with these observations, we found that the WT SHC014-CoV spike was almost completely resistant to neutralization by an RBD-directed antibody, whereas the mutants were highly sensitive (Fig 7E). To our knowledge, these dramatic shifts in neutralization susceptibility are much greater than any reported to date, and they attest to both the extremely closed nature of the WT SHC014-CoV spike and the degree to which the entry-enhancing substitutions reported in this study afford spike opening. Mutants of the type described herein may therefore suffer a considerable fitness cost in a physiological context. However, they may also provide an advantageous genetic background upon which to accrue additional mutations that maximize fitness by balancing spike stability, entry efficiency, and susceptibility to antibody neutralization during zoonotic adaptation of bat-origin CoVs.

## Methods

### Cell lines

293FT cells (Thermo Fisher) were cultured in high-glucose Dulbecco's Modified Eagle Medium (DMEM, Thermo Fisher) supplemented with 10% fetal bovine serum (FBS, Gemini),

1% Glutamax (Thermo Fisher), and 1% penicillin-streptomycin (P/S, Thermo Fisher). African green monkey kidney Vero Cells (ATCC) were cultured in high-glucose DMEM supplemented with 2% FBS, 1% Glutamax, and 1% P/S. Embryonic kidney fibroblast 293T cells (ATCC) were cultured in DMEM supplemented with 10% FBS, 1% Glutamax, and 1% P/S.

DBT-9 cells (gift of Ralph Baric, source unknown) [103,104] were cultured in Minimum Essential Medium α (MEMα, Thermo Fisher) supplemented with 10% FBS, 1% Glutamax, 1% P/S, and 1% Amphotericin (Thermo Fisher). Cells overexpressing ACE2 were also cultured with 5ug/mL puromycin (Gibco).

These cell lines were subcultured every 2–3 days using 0.05% Trypsin/EDTA (Gibco). All cell lines were maintained in a humidified 37°C incubator supplied with 5% $CO_2$.

## Plasmids

Plasmids encoding human codon-optimized spike (S) of SHC014-CoV (GenBank accession number KC881005.1), WIV-1-CoV (GenBank accession number KC881007), BtKY72-CoV (GenBank accession number KY352407), and RmYN02-CoV (GenBank accession number MW201982) in the VSV genome with an eGFP gene, were generated as previously described [42,43,45,46,105].

## Sarbecovirus sequence alignment

Sequence alignment of amino acid sequences of the fusion peptide proximal region (FPPR) of the S protein of selected sarbecoviruses were aligned using CLUSTAL Omega [106].

## Generation of recombinant VSVs

A plasmid encoding the VSV genome was modified to replace its glycoprotein, G, with the wild-type or mutant spike glycoprotein gene of SHC014-CoV, WIV-1-CoV, BtKY72-CoV, or RmYN02-CoV with a 21-amino acid C-tail truncation. The VSV genome also encodes for an eGFP reporter gene as a separate transcriptional unit. Replication-competent, recombinant VSVs (rVSVs) bearing these S proteins were generated via a plasmid-based rescue system in 293FT cells as previously described [42,43,45,46,105]. Briefly, 293FT cells were transfected with the VSV plasmid and plasmids expressing T7 polymerase and VSV N, P, M, G, and L proteins using polyethylenimine. Forty-eight hours post transfection, supernatants from the transfected cells were transferred to Vero cells. No exogenous trypsin was added to cell cultures. Rescues of SHC014-CoV S(A835D) and WIV-1-CoV S(WT) and S(A835D) were conducted at 37°C. BtKY72-CoV S(A835D) and RmYN02-CoV S(A806D) could be rescued at 32°C but not 37°C. Viral growth was monitored by an eGFP reporter every day. Viral stocks were plaque purified on Vero cells. Spike sequences were amplified from viral genomic RNA by RT-PCR and analyzed by Sanger sequencing and/or minION sequencing.

## Generation of pseudotyped VSVs

Single cycle VSV pseudotypes bearing SHC014-CoV S proteins and an eGFP reporter, were produced in 293T cells as described previously [107,108]. Briefly, 293T cells were transfected with a plasmid encoding an expression vector and the SHC014-CoV S. Two days later, cells were infected with a passage stock of VSVG/ΔG for 1h at 37°C. Cells were washed eight times with high glucose media to remove any residual VSV-G. Viral supernatant was harvested two days later and pelleted by ultracentrifugation.

## Generation of replication-competent delivery particles (RDPs)

Single-cycle SARS-CoV-2 RDPs bearing WT or mutant SHC014-CoV spikes were generated as described previously [48]. Briefly, BHK-21 cells were transfected with plasmids encoding the full-length WT or mutant SHC014-CoV spikes. The following day, cells were electroporated with ΔSpike nsp1- SARS-CoV-2-mNeonGreen replicon RNA and N mRNA. At 24h postelectroporation, the RDP-containing supernatants were collected and filtered through a 0.4-μM filter to remove cell debris.

## SHC014-CoV spike glycoprotein ectodomain expression and purification

Amino acids 1–1191 of SHC014-CoV spike were coupled to a C–terminal foldon motif, 3C protease site, 8X histidine tag, and Twin-Strep tag and cloned into a pαH vector. For variants, substitutions F294L and A835D were incorporated into this background. All variants were expressed by polyethylenimine-induced transient transfection of FreeStyle 293-F cells (Thermo Fisher). After 6 days, the cell supernatants were harvested by centrifugation and clarified via passage through a 0.22 μm filter. Variants were purified from filtered supernatant via gravity flow over StrepTactin resin (IBA) followed by gel-filtration chromatography on a Superose 6 10/300 column (GE Healthcare) into a buffer consisting of 2 mM Tris pH 8.0, 200 mM NaCl, 0.02% NaN$_3$.

## *HsACE2* expression and purification

Amino acids 1–615 of HsACE2 were coupled to a PreScission site, 8X histidine tag, and Strep Tag II and cloned into a pαH vector. HsACE2 was expressed by polyethylenimine-induced transient transfection of FreeStyle 293-F cells (Thermo Fisher). After 6 days, the cell supernatant was harvested by centrifugation and clarified via passage through a 0.22 μm filter. *Hs*ACE2 was then purified from filtered supernatant via gravity flow over StrepTactin XT resin (IBA) followed by gel-filtration chromatography on a Superdex 200 Increase 10/300 column (GE Healthcare) into a buffer consisting of 2 mM Tris pH 8.0, 200 mM NaCl, 0.02% NaN$_3$.

## Nanopore sequencing

Viral RNA was isolated from rescue population supernatants (Zymogen Quick-RNA viral Kit). cDNA was then generated through reverse transcription (Invitrogen Superscript III). PCR reactions were performed to amplify the cDNA population (NEB Q5 Hotstart). Following this Oxford Nanopore, MinION long read DNA sequencing was performed to identify mutations present in the viral population. Sequenced bases were called by Guppy (https://community.nanoporetech.com/docs/prepare/library_prep_protocols/Guppy-protocol/v/gpb_2003_v1_revax_14dec2018/guppy-software-overview) and the resulting sequences were assembled using the de novo assembly tool Shasta (https://github.com/chanzuckerberg/shasta/blob/master/docs/ComputationalMethods.html). The resulting assembled contigs were aligned to the reference genome utilizing Smith-Waterman local alignments.

## Detection of SHC014-CoV spike surface expression

293FT cells were seeded in six-well plates. 24 hours later, cells were transfected with 2ug of expression plasmids encoding either nothing, SHC014-CoV S WT, or A835D, and 1ug of a plasmid encoding for RFP. 24 hours post transfection, cells were blocked with 0.5% bovine serum albumin (BSA, Sigma Aldrich) in PBS for 30 minutes at 4˚C. SHC014-CoV S was stained by a spike-specific mAb, S309 (20ug/mL) followed by anti-human Alexa Fluor 488

(4ug/mL) for 1 hour at 4˚C each. After extensive washing, stained cells were filtered through a 41-μm nylon net filter (Millipore) and analyzed using a BD FACSCalibur Flow Cytometer and FlowJo software.

### Detection of SHC014-CoV spike total expression

293FT cells were seeded in six-well plates. 24 hours later, cells were transfected with 2ug of expression plasmids encoding either nothing, SHC014-CoV S WT, or A835D, and 1ug of a plasmid encoding for RFP. 24 hours post transfection, cells were collected at fixed with 4% formaldehyde at room temperature (RT) for 5 minutes. Cells were washed and permeabilized with 1X permeabilization buffer (Tonbo Biosciences) for 15 minutes at RT. Cells were blocked with 10% FBS in permeabilization buffer for 30 minutes at RT. Following washing, cells were stained with S309 (20ug/mL) followed by anti-human Alexa Fluor 488 (4ug/mL). After extensive washing, stained cells were filtered through a 41-μm nylon net filter (Millipore) and analyzed using a BD FACSCalibur Flow Cytometer and FlowJo software.

### ELISA for assessment of spike incorporation into scVSV particles

High-protein binding 96-well ELISA plates (Corning) were coated overnight at 4˚C with scVSV-SHC014-CoV particles bearing WT or A835D mutant spikes. Plates were blocked with 3% PBSA for 1h at 37˚C and particles were subsequently probed with 100nM of S309 or anti-VSV M mAb (1:1000 dilution) for 1h at 37˚C. Following washing, anti-human IgG (for S309) or anti-mouse IgG (for anti-VSV M mAb) secondary antibody conjugated to horseradish peroxidase was added for 1h at 37˚C. Plates were washed, 1-Step Ultra TMB-ELISA Substrate Solution (Thermo Fisher) was added and quenched with the addition of 2 M sulfuric acid. Absorbance was read at 450 nm. The ratio of spike-to-VSV M was calculated for evaluation of spike incorporation per scVSV particle.

### ELISA for detection of scVSV-SHC014-CoV S binding to soluble *HsACE2*

High-protein binding 96-well ELISA plates (Corning) were coated with 2ug/mL of soluble HsACE2 overnight at 4˚C. Plates were then washed with PBS and blocked with PBS containing 3% BSA for 1h at 37˚C. Blocked plates were exposed to scVSV-SHC014-CoV, in duplicate, at pre-titrated amounts that were diluted in a 3-fold serial dilution, for 1 hour at 37˚C. Plates were washed with PBS and incubated with 10nM of ADG-2 for 1h at 37˚C. Following washing, anti-human IgG secondary antibody conjugated to horseradish peroxidase was added for 1h at 37˚C. Plates were washed, 1-Step Ultra TMB-ELISA Substrate Solution (Thermo Fisher) was added and quenched with the addition of 2 M sulfuric acid. Absorbance was read at 450 nm.

### ELISA for detection of scVSV-SHC014-CoV S binding to antibodies

High-protein binding 96-well ELISA plates (Corning) were coated with genome normalized amounts of scVSV-SHC014-CoV particles bearing WT or A835D mutant spikes. Plates were blocked with 3% PBSA for 1h at 37˚C and particles were subsequently probed with 100nM of S309, ADG-2, S2-21, SARS-CoV-2 Spike S2 rabbit polyclonal antibody (Spike S2 PAb) (https://www.sinobiological.com/antibodies/cov-spike-40590-t62) (1:5000 dilution), or anti-VSV M mAb (1:1000 dilution) for 1h at 37˚C. Following washing, anti-human IgG (for S309, ADG-2, and S2-21), anti-mouse IgG (for anti-VSV M mAb), or anti-rabbit IgG (for SARS--CoV-2 Spike S2 rabbit polyclonal antibody) secondary antibody conjugated to horseradish peroxidase was added for 1h at 37˚C. Plates were washed, 1-Step Ultra TMB-ELISA Substrate

Solution (Thermo Fisher) was added and quenched with the addition of 2 M sulfuric acid. Absorbance was read at 450 nm.

## Biolayer interferometry (BLI)

An IgG targeting T4 fibritin (foldon) was immobilized to anti-human Fc (AHC) Octet biosensors (FortéBio) to a response level of 1.0 nm. Tips were then submerged into solutions of SHC014-CoV spike variants in 20 mM Tris pH 8.0, 150 mM NaCl, 1 mg/mL bovine serum albumin, and 0.01% Tween-20, capturing variants to similar response levels. Spike-bound biosensors were subsequently dipped into 500 nM HsACEII in 20 mM Tris pH 8.0, 150 mM NaCl, 1 mg/mL bovine serum albumin, and 0.01% Tween-20 [109] to observe receptor association. Dissociation of HsACEII was then observed by submersion into a buffer consisting of 20 mM Tris pH 8.0, 150 mM NaCl, 1 mg/mL bovine serum albumin, and 0.01% Tween-20. The relative proportion of accessible RBDs was quantified as previously described [109]. Data were collected in triplicate.

## Cryo-EM data collection

The SHC014-CoV S ectodomain data set was collected on a Thermo Scientific Glacios operating at 200 kV utilizing a Falcon 4 direct electron detector. C-Flat 1.2/1.3 grids (Electron Microscopy Sciences) were glow discharged at 20 mA for 30 seconds. Samples were prepared by application of 4 uL of a 1.0 mg/mL solution of spike onto grids. Grids were then blotted for 4 seconds using a blot force of -3 and a wait time of 5 seconds on a Vitrobot Mark IV (Thermo Scientific) set to 100% humidity at room temperature. The blotted grid was subsequently plunge-frozen into liquid ethane. All data were collected at 150,000x magnification, which gave a pixel size of 0.94 Å. In total, 2,025 exposures were taken using a defocus range of -1 to -2 microns, an exposure time of 13.2 seconds, and a total electron exposure of ~50 e$^-$/Å$^2$. Data were collected using SerialEM.

## Cryo-EM data processing

The cryo-EM data were pre-processed using cryoSPARC Live v3 [110] for motion correction and patch CTF estimation. Exposures were subsequently transferred to cryoSPARC v3 for manual curation, blob particle picking, particle extraction, and 2D classification. 1,653 exposures were accepted after curation, which resulted in 649,312 particle picks. After 2D classification, 150,716 particles remained and were used in a 4-class *ab initio* reconstruction job. The four output volumes were used with all particles for heterogeneous refinement. The best class contained 79,540 particles, which was used for homogenous refinement with both C1 and C3 symmetry, leading to reconstructions at 3.7 Å and 3.2 Å resolution, respectively. A total of 76,442 particles were then subjected to a non-uniform refinement with C3 symmetry, leading to a 3.1 Å resolution reconstruction.

An initial model of the WT SHC014-CoV spike was generated using AlphaFold2 [61] and fitting into the 3.1 Å cryo-EM map in ChimeraX [111,112]. The model was subsequently refined iteratively in Isolde [113], Coot [114], and Phenix [115]. Structural figures were generated in ChimeraX.

## *In silico* binding free energy calculation between spike protomers

Optimization of the cryo-EM structure: Missing loops were modeled via various tools including Swiss Model [116], AlphaFold2 [61], Prime Homology Modeling [117], and Modeller [118]. Resultant loops were grafted onto the resolved Cryo-EM structure via in-house Visual

Molecular Dynamics scripts. Uniprot and comparison to human SARS-CoV-2 spike system structure were used to identify necessary disulfide bonds (15 per monomer). Ectodomain glycosites were predicted and confirmed as described [119,120]. Our modeled construct is fully N-/O-glycosylated following a similar human glycoprofile as described [109], which was consistent with Watanabe and co-workers [121]. Protonation states for all titratable residues were assigned using PROPKA [122] at pH 7.4, and histidine protonation states were assigned with Schrodinger's Protein Preparation Wizard [123]. The resultant SHC014-CoV glycoprotein was parameterized for minimization and short equilibration according to CHARMM36 all-atom additive force fields for proteins and glycans [124,125]. The SHC014-CoV construct was fully solvated with explicit water molecules using the TIP3P model [126] and neutralized with a 150 mM concentration of sodium and chloride ions. The system was then minimized, heated to 310 K, and briefly equilibrated using NAMD3 [127] with standard molecular dynamics simulations protocols. For a complete set of grafting, system construction, glycoprofile details, minimization, heating, equilibration scripts, and complete methodological details, please see all shared scripts and shared model files which will be provided on the AmaroLab website (https://amarolab.ucsd.edu/covid19.php).

Binding free energy calculation (interface ΔΔG): We ran two separate methods (FoldX and flex ddG) to estimate the binding affinity change between interacting protomers in the SCH014 spike trimer [128,129]. Using FoldX, we performed five rounds of internal structure optimization (FoldX repairPDB), and iteratively modeled and assessed A835D (25 iterations; FoldX BuildModel and AnalyseComplex). Flex ddG: calculation parameters were set as follows, nrstruct = 35, max_minimization_iter = 5000, abs_score_convergence_thresh = 1m, number_backrub_trials = 35000. In order to make the calculation tractable with flex ddG, we assessed the effect of A835D using a subset of residues (chain A: residues 691 to 1124; chain C: residues 18 to 689). The reported ΔΔG for each method corresponds to the averaged ΔΔG.

## Virus infection assays

24 hours prior to infection, adherent cells were seeded in high glucose media on 96 well plates (Corning). Vero cells were plated at $2 \times 10^4$ cells per well. DBT-9 cells were plated at $1.8 \times 10^4$ cells per well. Prior to infection, virus was diluted in corresponding media and infected cells were incubated at 37°C for 12–18 hours. Cells were then fixed with 4% paraformaldehyde, stained with Hoechst-33342 (Invitrogen) before eGFP+ cells were counted using a Cytation 5 reader (BioTek Instruments). Infection levels were calculated as a percentage of eGFP+ cells over the total number of cells.

## Temperature-dependent infection assays

DBT-9 cells overexpressing *Ra*ACE2 were seeded 24 hours prior to infection in high glucose media on 96 well plates at $1.8 \times 10^4$ cells per well. scVSV-SHC014-CoV particles were diluted in PBS and incubated at a temperature range from 37°C to 55°C (actual range per virus indicated in Results) for 1 hour and subsequently placed on ice. After cooling, virus was added to cells and incubated at 37°C for 12–18 hours. Cells were then fixed with 4% paraformaldehyde, stained with Hoechst 33342 (Invitrogen) before eGFP+ cells were counted using a Cytation 5 reader (BioTek Instruments). Infection levels were calculated as a percentage of eGFP+ cells over the total number of cells.

## Temperature-dependent binding assays

scVSV-SHC014-CoV particles were diluted in PBS and incubated at a temperature range from 37°C to 55°C (actual range per virus indicated in Results) for 1 hour and subsequently placed

on ice. After cooling, virus was added onto high-binding 96-well plates precoated with soluble *HsACE2*. Plates were blocked with 3% BSA in PBS. SHC014-CoV S was detected with ADG-2. Bound antibody was then detected with an anti-human antibody conjugated to HRP. Plates were washed, 1-Step Ultra TMB-ELISA Substrate Solution (Thermo Fisher) was added and quenched with the addition of 2 M sulfuric acid, per manufacturer recommendations. Absorbance was read at 450 nm. All binding steps were carried out at 37˚C for 1h. Binding curves were generated using Prism (GraphPad Software, La Jolla, CA).

## ADG-2 neutralization assay

mAb ADG-2 was serially diluted and mixed with pre-diluted scVSV-SHC014-CoV WT, A835D, F294L, or F294L A835D in infection media (DMEM, 2% FBS, 1% P/S, 1% Q). Mixtures were incubated for 1 hr at 37˚C. Virus/antibody inoculum was added to DBT-*Ra*ACE2 cells, pre-seeded in 96 well plates and incubated for 12–18 hours. Cells were fixed with 4% paraformaldehyde (Sigma), washed with PBS, and stored in PBS containing Hoechst-33342 (Invitrogen) at a dilution of 1:4000. Viral infectivity was measured by automated enumeration of GFP-positive cells from captured images using a Cytation5 automated fluorescence microscope (BioTek) and analyzed using the Gen5 data analysis software (BioTek).

## Linear model for non-additive mutant effects

Infectivity data on DBT-*Hs*ACE2 cells and *Hs*ACE2 binding ELISA data for scVSVs bearing WT, F294L, A835D, or F294L+A835D SHC014-CoV spikes were analyzed according to the simplex regression method of Miton and colleagues to extract additive and non-additive mutational effects, essentially as described [66]. Log-transformed replicate infectivity and binding ELISA data were calculated for each genotype, and mutant–WT values were processed with python script 'simplex_regression.py' available at https://github.com/danderson8/Epistasis to determine the best-fit linear model and extract the contribution of each substitution to the phenotype of the double-mutant. See S2 Table Legend for more details.

## Statistical analysis

Statistical details for each experiment, including number of replicates (n) and types of statistical tests used, are reported in respective figure legends. Statistical analysis was carried out using GraphPad Prism (V.10).

## Supporting information

**S1 Fig. rVSV-WIV-1-CoV WT and A835D infection curves on DBT-9 cells overexpressing ACE2 orthologs.** Parental DBT-9 cells or DBT-9 cells overexpressing *Hs*ACE2 or *Ra*ACE2 were infected with rVSV-WIV-1-CoV WT or A835D particles. Infection was scored by eGFP expression at 12 hours post-infection (average±SD, n = 2–5 from 3 independent experiments). Groups (WT vs. mutant for each cell line) were compared with Welch's t-test with Holm-Šídák correction for multiple comparisons, ns $p > 0.05$; ** $p < 0.01$; *** $p < 0.001$; **** $p < 0.0001$. Only the statistically significant comparisons are shown. (TIF)

**S2 Fig. A835D phenotype is independent of spike incorporation into VSV particles. (a)** 293FT cells were transfected with plasmids expressing WT or A835D SHC014-CoV spike and immunostained for total protein expression (left) or cell surface expression (right) 24 hours post transfection by a spike-specific antibody and analyzed using flow cytometry (average±SD, n = 3). Groups were compared by one-way ANOVA. **(b)** scVSV-SHC014-CoV S WT or

A835D particles were coated on ELISA plates and probed with a spike-specific mAb or VSV M-specific mAb. The spike-to-M ratio demonstrates VSV particle-specific spike incorporation (average±SD, n = 6 from 3 independent experiments). Groups were compared by one-way ANOVA. **(c)** Vero cells were infected with pre-titrated amounts of scVSV-SHC014-CoV S WT or A835D. Infection was scored by eGFP expression at 16–18 hours post-infection (average ±SD, n = 10–11 from 3 independent experiments). A range of $3.18×10^1$ to $1.24×10^7$ viral GEQ was used. Groups were compared with Welch's t-test with Holm-Šídák correction for multiple comparisons. ns p>0.05; ** p<0.01; *** p<0.001; **** p<0.0001. In panel c, only the statistically significant comparisons between WT and A835D are shown.
(TIF)

**S3 Fig. Binding to panel of antibodies by scVSV-SHC014 WT and A835D reveal allosteric changes in spike structure.** Genome normalized amounts of scVSV-SHC014-CoV S WT and A835D were coated on an ELISA plate and detected with a panel of antibodies, followed by HRP-conjugated secondary antibody. $1.6×10^6$ viral GEQ was used per well. Groups (WT vs. mutant) were compared with two-way ANOVA with Tukey's correction for multiple comparisons, ns p>0.05; ** p<0.01; *** p<0.001; **** p<0.0001. Only the statistically significant comparisons are shown.
(TIF)

**S4 Fig. Cryo-EM processing pipeline.** Cryo-EM processing workflow for the WT SHC014-CoV spike. The indicated steps were conducted using cryoSPARC v3: motion correction, contrast transfer function (CTF) estimation, 2D classification, heterogenous refinement, homogenous refinement with C3 symmetry imposed and non-uniform refinement with C3 symmetry imposed. Scale bars in micrograph and 2D classes represent 10 nm.
(TIF)

**S5 Fig. Cryo-EM map-to-model fit.** Fit of the SHC014-CoV S model to the refined cryo-EM map. **(a)** Global map-to-model fit shown with one protomer hidden for clarity. **(b)** Global map-to-model fit z-slice with all protomers shown. **(c)** Local fit with side chains shown surrounding Phe294, with Lys266, Lys288, and Phe294 labelled. **(d)** Local fit with side chains shown surrounding Ala835, with Val556, Ala835, and Val946 labelled. For all panels, a threshold value of 0.19 was used to visualize the cryo-EM map.
(TIF)

**S6 Fig. scVSV-SHC014-CoV particles bearing A835E spikes have enhanced infectivity compared to WT.** Parental DBT-9 cells or DBT-9 cells overexpressing *Hs*ACE2 or *Ra*ACE2 were infected with pre-titrated amounts of scVSV-SHC014-CoV particles bearing WT, A835D, or A835E spike. Infection was scored by eGFP expression at 16–18 hours post-infection (average±95%CI, n = 4–8 from 2–3 independent experiments). A range of $4.6×10^2$ to $1.0×10^6$ viral GEQ was used. Groups (WT vs. mutant for each cell line) were compared with Welch's t-test with Holm-Šídák correction for multiple comparisons. ns p>0.05; ** p<0.01; *** p<0.001; **** p<0.0001. Only the statistically significant comparisons are shown.
(TIF)

**S7 Fig. Comparison of WT and mutant SHC014-CoV S ectodomains. (a)** Overlaid size-exclusion chromatography $Abs_{280nm}$ traces show relative protein yield. WT, A835D, F284L, and F294L+A835D SHC014-CoV spikes were expressed and purified in tandem from the same cell line in the same culture volume. Area under the curve represents protein expression. A835D-containing variants exhibit a large decrease in protein expression relative to the WT and F294L SHC014-CoV spikes. **(b)** WT SHC014-CoV S visualized in a cryo-EM micrograph

embedded in the vitreous ice. **(c)** A835D SHC014-CoV visualized in a cryo-EM micrograph, prepared using the same protocol as in (a). Both spike proteins were expressed and purified at the same time using the same procedures. Scale bars in both represent 10 nm.
(TIF)

**S8 Fig. Presence of T842A mutation does not change F294L or F294L+A835D infectivity phenotype.** Parental DBT-9 cells or DBT-9 cells overexpressing *Hs*ACE2 or *Ra*ACE2 were infected with pre-titrated amounts of scVSV-SHC014-CoV particles bearing WT, A835D, F294L, A835D+T842A, A835D+F294L, or F294L+A835D+T842A spike. Infection was scored by eGFP expression at 16–18 hours post-infection (average±95%CI, n = 3–9 from 2–3 independent experiments). A range of $7.15\times10^2$ to $4.7\times10^6$ viral GEQ was used. Groups (WT vs. mutant for each cell line) were compared with Welch's t-test with Holm-Šídák correction for multiple comparisons. ns p>0.05; ** p<0.01; *** p<0.001; **** p<0.0001. Only the statistically significant comparisons are shown.
(TIF)

**S9 Fig. Location and interactions of NTD substitution in SHC014-CoV spike. (a)** The aromatic ring of Phe294 is positioned between Lys266 and Lys288. **(b)** The distance between Phe294 and the RBD is depicted, with NTD residues and residues after SD1 hidden for clarity. Labelled residues are modeled as spheres. **(c)** Alignment of amino acid sequences in the F294 region (rounded rectangle) for selected coronavirus spike proteins. Subgenera are indicated in italics. Sarbecoviruses are color-coded by clade (1a: SARS-CoV–like, red; 1b: SARS-CoV-2–like, green; 2: Southeast Asian bat-origin CoV, blue; 3: non-Asian bat-origin CoV, purple). Spikes investigated in the current study are in bold.
(TIF)

**S1 Table. EM data collection, processing and refinement statistics.**
(DOCX)

**S2 Table. Modeling non-additive contributions of F294L and A835D to scVSV-SHC014-CoV fitness.** The simplex regression method of Miton and colleagues was used to estimate the additive and non-additive contributions of each mutation to the double-mutant phenotype. Tabs 1–3: Analysis of scVSV:*Hs*ACE binding. Tabs 4–6: Analysis of scVSV:DBT-*Hs*ACE infectivity. Tabs 1, 4: Log-transformed and WT-subtracted data. Tabs 2, 5: Best-fit model for each dataset. 'order 1' = additive contributions. Higher-order (i.e., epistatic) contributions did not reach statistical significance (p = 0.05). Tabs 3, 6: Effect of each mutation position.
(XLSX)

**S1 File. Nanopore sequencing of rVSV-SHC014-CoV reveals A835D mutation.** Consensus sequence of spike gene in the rescued rVSV-SHC014-CoV viral population generated by assembly of nanopore sequencing reads. A single point mutation encoding the substitution A835D was observed. Sequence provided in plain-text file in fasta format.
(FA)

**S2 File. Nanopore sequencing of rVSV-SHC014-CoV reveals F294L mutation.** Consensus sequence of spike gene in the rescued rVSV-SHC014-CoV viral population generated by assembly of nanopore sequencing reads. The starting rescue plasmid encoded the A835D and T842A substitutions. A single additional point mutation encoding the substitution F294L was observed. Plain-text file with sequence in fasta format.
(FASTA)

## Acknowledgments

We thank J. Janer, K. Paez, E. Valencia, and M. Ramirez (Einstein) for laboratory management and technical support. We thank Ralph S. Baric (R.S.B.) for his provision of DBT-9–based cell lines and R.S.B. and Michael Letko suggestions on a preliminary version of this manuscript. We also thank Jimmy Gollihar for providing the S2-directed monoclonal antibody, S2-21.

## Author Contributions

**Conceptualization:** Alexandra L. Tse, Cory M. Acreman, Rommie E. Amaro, Rohit K. Jangra, Jason S. McLellan, Kartik Chandran, Emily Happy Miller.

**Investigation:** Alexandra L. Tse, Cory M. Acreman, Inna Ricardo-Lax, Jacob Berrigan, Gorka Lasso, Toheeb Balogun, Fiona L. Kearns, Lorenzo Casalino, Georgia L. McClain, Amartya Mudry Chandran, Charlotte Lemeunier, Rohit K. Jangra, Emily Happy Miller.

**Methodology:** Alexandra L. Tse, Cory M. Acreman, Inna Ricardo-Lax, Jacob Berrigan, Gorka Lasso, Rommie E. Amaro, Charles M. Rice, Rohit K. Jangra, Jason S. McLellan, Kartik Chandran, Emily Happy Miller.

**Supervision:** Rommie E. Amaro, Charles M. Rice, Jason S. McLellan, Kartik Chandran, Emily Happy Miller.

**Visualization:** Alexandra L. Tse, Cory M. Acreman, Gorka Lasso, Jason S. McLellan, Kartik Chandran, Emily Happy Miller.

**Writing – original draft:** Alexandra L. Tse, Cory M. Acreman, Gorka Lasso, Jason S. McLellan, Kartik Chandran, Emily Happy Miller.

**Writing – review & editing:** Alexandra L. Tse, Cory M. Acreman, Inna Ricardo-Lax, Jacob Berrigan, Gorka Lasso, Toheeb Balogun, Fiona L. Kearns, Lorenzo Casalino, Georgia L. McClain, Amartya Mudry Chandran, Charlotte Lemeunier, Rommie E. Amaro, Charles M. Rice, Rohit K. Jangra, Jason S. McLellan, Kartik Chandran, Emily Happy Miller.

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
