## [Decision Letter · Decision Letter 0]

18 Aug 2024

Dear Dr. Chandran,

Thank you very much for submitting your manuscript "Distinct pathway for evolution of enhanced receptor binding and cell entry in SARS-like bat coronaviruses" for consideration at PLOS Pathogens. As with all papers reviewed by the journal, your manuscript was reviewed by members of the editorial board and by several independent reviewers. The reviewers appreciated the attention to an important topic. Based on the reviews, we are likely to accept this manuscript for publication, providing that you modify the manuscript according to the review recommendations.

Your manuscript has been reviewed by three expert reviewers, all of whom agreed it was an interesting and high-quality study.

All three reviewers made constructive and perceptive comments. If you address the comments thoroughly, the revised manuscript might be approved after editorial review without sending for re-review. Specifically:

- Please address all comments by Reviewer 1.

- Please also address all comments by Reviewer 2, except of experiments with additional antibodies beyond ADG2 that are at your discretion. As Reviewer 3 also notes, this would strengthen the manuscript.

- Reviewer 3 makes numerous perceptive comments, although some of them require substantial work that may exceed the scope of the original study. So address as many as you can with new data, but some can be addressed with additional thorough textual discussion instead if needed. You do definitely need to address the comment about providing more details about generating BtKY72 and RmYN02 with the added mutation though. You do not need to experimentally address the comment about using a SHC014-backbone based replicon system, as that would be a massive amount of additional work.

- Also, as alluded by Reviewer 2, please add a bit more discussion of studies of similar phenomenon in SARS-CoV-2 evolution, and how it relates to what you observe. In particular, there is clear experimental evidence that SARS-CoV-2 spike has fixed mutations that have modulate RBD conformation during its evolution in humans, and also that mutations that alter RBD conformation affect serum antibody neutralization. See for instance Zhang et al, Molecular Cell, 84:2747 and Dadonaite et al, Nature, 631:617.

- Following Reviewer 3 request, please clarify or improve description of the virus classification and naming frameworks used in the manuscript (see Editors comments below).

Sincerely,

Jesse D Bloom, Ph.D.

Guest Editor

PLOS Pathogens

Alexander Gorbalenya

Section Editor

PLOS Pathogens

Michael Malim

Editor-in-Chief

PLOS Pathogens

orcid.org/0000-0002-7699-2064

Editor's comments:

- Please address all comments by Reviewer 1.

- Please also address all comments by Reviewer 2, except I will not require you to perform experiments with additional antibodies beyond ADG2. As Reviewer 3 also notes, this would strengthen the manuscript but I don't consider that new work essential for acceptance. So do it if you can (it would strengthen paper), but I won't require it.

- Reviewer 3 makes numerous perceptive comments, although some of them require substantial work that may exceed the scope of the original study. So address as many as you can with new data, but some can be addressed with additional thorough textual discussion instead if needed. You do definitely need to address the comment about providing more details about generating BtKY72 and RmYN02 with the added mutation though. You do not need to experimentally address the comment about using a SHC014-backbone based replicon system, as that would be a massive amount of additional work.

- I also suggest (as also alluded by Reviewer 2) that you add a bit more discussion of studies of similar phenomenon in SARS-CoV-2 evolution, and how it relates to what you observe. In particular, there is clear experimental evidence that SARS-CoV-2 spike has fixed mutations that have modulate RBD conformation during its evolution in humans, and also that mutations that alter RBD conformation affect serum antibody neutralization. See for instance Zhang et al, Molecular Cell, 84:2747 and Dadonaite et al, Nature, 631:617.

- Please introduce subgenus and genus taxa, according to virus taxonomy rules, e.g. replace "...isolated seven distinct full-length CoV sequences belonging to the subgenus of SARS-like betacoronaviruses (sarbecoviruses), including two from novel agents— Rs3367 and RsSHC014—and determined their full-length genome sequences" with "...determined full-length genomic sequences of seven CoVs belonging to the subgenus *Sarbecovirus*, genus *Betacoronavirus*, including two from novel agents — Rs3367 and RsSHC014."

- Please introduce phylogenetic clades used in Fig. 1 and elsewhere; note, they may be prone to intra-species homologous recombination.

Reviewer Comments (if any, and for reference):

Reviewer's Responses to Questions

**Part I - Summary**

Reviewer #1: With a primary focus on the bat RsSHC014 sarbecovirus, Tse et al. discover spike mutations that enhance the propensity of RBDs to sample open conformations and thereby enhance cell entry by increasing accessibility of receptor engagement. This points to an important mechanism of restricting or enabling cross-species transmission for animal coronaviruses whose RBDs may already bind a novel host receptor but are restricted in entry due to spike dynamics. I find the study nicely conducted and nicely presented. I have just two comments below, neither of which detract from the overall study quality.

Reviewer #2: The manuscript titled “Distinct pathway for evolution of enhanced receptor binding and cell entry in SARS-like bat coronaviruses” by Tse, Ackerman et al. utilize a comprehensive toolbox of technologies covering virology, structural biology and biochemistry to uncover the effect of an FPPR mutation in a bat coronavirus spike protein on its RBD exposure and successful rescue of replication-competent rVSVs bearing the mutant spike, whereas, the WT could not be recovered despite several attempts. This mutation – A835D – was first identified in the SHC014-CoV S and then shown to be consistent in its effect when incorporated at analogous positions in different bat Coronaviruses. The structural studies were performed on the WT spike and revealed a closed spike conformation, a finding consistent with receptor utilization, entry capabilities and neutralization data. A role for the A835D mutation in destabilizing local contacts was proposed based on structural modeling of the mutation on the WT structure. The use of AlphaFold based structure prediction to glean structural insights and design virological experiments based on these, while the experimental cryo-EM structures were in progress, was seen as a plus. Indeed, these investigations led to the identification of an NTD mutation, F294L, that showed synergistic effect with the A835D mutation to enhance RBD exposure. Overall, a robust study using multiple synergistic methodologies and a well-written easy to read paper.

Reviewer #3: In this study, Tse et al. used recombinant vesicular stomatitis viruses (rVSVs) and an authentic coronavirus-like vector to show that mutations (A835D, F294L) in the bat SARSr-CoV SHC014 spike, affect viral cell entry. They also found that the A835 residue in the S2 subunit is conserved across bat SARSr-CoV spikes, and the A835D mutation can facilitate the rescue of rVSVs bearing spikes from clade 2 and 3 bat sarbecoviruses. The authors further investigated the underlying mechanism and found that the enhancement of viral entry is due to a structural change in the SHC014 spike RBD, shifting it from a "down" to an "up" state, thereby increasing the availability of spike: receptor ACE2 interaction. Overall, this study demonstrates that the different regions of the sarbecovirus spike (NTD, RBD, and S2 subunit) work synergistically in facilitating viral entry, rather than independently. This finding is important and meaningful for advancing our understanding of sarbecovirus cell entry mechanisms. It also offers new insights and hypotheses regarding the genetic evolution of sarbecoviruses, whether through natural selection or maybe host-to-host transmission. Understanding these molecular features and evolutionary pathways in sarbecovirus spikes is crucial for both tracking viral evolution and developing effective countermeasures.

**Part II – Major Issues: Key Experiments Required for Acceptance**

Reviewer #1: 1. Throughout the manuscript, the authors discuss epistasis and/or synergy between two mutations that enhance RBD opening (A835D and F294L). They correctly define positive epistasis as “a phenomenon in which the combined effect of two or more mutations in a protein is greater than (and/or distinct from) the sum of their individual effects”, but I’m not sure they apply this proper definition to the interpretation of their data. The current metrics for evaluating epistasis are a bit more qualitative than quantitative (i.e. visual comparison of plots in Figure 6) which complicates the identification of epistasis as a quantitative deviation from additive expectation, but for example in Figure. 6b, I do not see clear evidence of epistasis: F294L enhances viral infectivity, A835D enhances viral infectivity, and the double mutant has even further enhanced infectivity – as would be expected if the two mutations were additive. To truly prove synergistic epistasis, the authors would need (1) to define a quantitative metric to summarize each genotype’s phenotype; (2) prove the background scale on which mutations are expected to combine additively (e.g. addition of a mutation with effect +1 and another with +2 yields expected double mutant phenotype of +3 on a linearly additive scale (so epistasis is present if phenotype != +3), but some phenotypes are multiplicative e.g. a mutation with a 1.1-fold effect and another with a 1.2-fold effect would be expected to have a 1.32-fold effect when combined; and yet other phenotypes are simply nonlinear even in the absence of epistasis, e.g. diminishing returns epistasis), and (3) show that the double mutant deviates from the non-epistatic expectation. Synergy means that there is more increase in infectivity than expected from single mutant measurements, not simply that the double mutant has stronger infectivity than either single mutant alone. In fact, the authors own language points to the non-epistatic relationship between these two mutations’ effects, p. 17: “F294L+A835D conferred a striking **additive** enhancement in infectivity relative to the single mutants”.

Reviewer #2: (No Response)

Reviewer #3: 1. The bat SARSr-CoV SHC014 was initially identified from Rhinolophus sinicus. Why the authors didn’t use Rhinolophus affinis ACE2 in their study? The authors showed that the A835D mutation occurred during the rescue in Vero cells, but subsequent receptor usage assessments and measurements were conducted in DBT-human ACE2 and DBT-Rhinolophus affinis ACE2 cell lines, or with human and Rhinolophus affinis ACE2 proteins. The authors should also test the natural host ACE2 to determine if the mutations are compatible with its host receptor, including binding and infection assays, to assess whether the mutations enhance or diminish its own receptor interaction.

2. The authors demonstrate “we could recover rVSV-RmYN02-CoV S(A806D) and rVSV-BtKY72-CoV S(A837D)”, but no data is presented. Additionally, since clade 2 SARSr-CoV requires trypsin for recovery, what conditions were used for rVSV-RmYN02-CoV S(A806D) and rVSV-BtKY72-CoV S recovery? Was trypsin added? The authors should clarify these details so that readers can understand whether the mutation is the critical factor.

3. The authors showed that mutations in the SHC014 spike could shift its RBD from a "down" to an "up" state. Have they tested whether the same mutations in RmYN02-CoV S and BtKY72-CoV S would result in similar structural changes? If structural analysis is challenging, the authors could instead assess their neutralization abilities using different antibodies that recognize different epitopes, as they did with the SHC014 spike. Additionally, if these mutations in RmYN02-CoV S and BtKY72-CoV S also expose their RBD to an "up" state, would they be able to bind human ACE2? This is particularly relevant for BtKY72-CoV S, which naturally binds to its host's ACE2. With these mutations, would their binding affinity to their natural host's ACE2 and human ACE2 improve or decline?

4. Why did the authors use the SARS-CoV-2 backbone instead of the SHC014 backbone for the replicon-based system? Would the SHC014 backbone result in better or worse growth with its own spike? The authors should explore this further.

5. The authors used DBT-RaACE2 cells for the spike thermostability and temperature dependence experiments, while human ACE2 was used for binding experiments. Did the authors also use human ACE2 cells and RaACE2 protein for these tests? If so, are the results consistent? The authors should include these results in the manuscript.

6. The F294L mutation was observed in the SHC014 spike along with the A835D and T842A double mutations. Why did the authors test F294L only with A835D or alone, but not with both A835D and T842A mutations together?

7. In Fig. 6b, the F294L+A835D double mutations infect DBT-ACE2 null cells at higher virus concentration, how to interpret this result? Does it indicate that the SHC014 spike can enter cells independently of ACE2 with these two mutations?

8. The authors tested the antibody sensitivity of the SHC014 spike with mutations using only one antibody that binds to the RBD 'Up' state. They should also test antibodies that specifically recognize the RBD 'Down' state to determine if the mutated spike is less sensitive compared to the WT with these antibodies.

9. The authors demonstrated that the A835D, F294L, or double mutations in the SHC014 spike enhance human ACE2 binding and increase infectivity in DBT-human ACE2 and Ra ACE2 cell lines. Since infection also involves membrane fusion and A835D is located in the S2 subunit, the authors should investigate whether this mutation affects membrane fusion efficiency, such as changes in protease dependence or fusion ability.

10. In Fig. S2, did the authors use a specific SARS-CoV-2 spike antibody to measure scVSV-SHC014-CoV S?

**Part III – Minor Issues: Editorial and Data Presentation Modifications**

Reviewer #1: 2. The authors do discuss the T372A mutation underlying SARS-CoV-2 emergence in the discussion, but I found its absence notable in the introduction. This mutation has been shown to enhance replication and suggested to underlie SARS-CoV-2’s propensity for RBD opening relative to bat sarbecoviruses like RsSHC014 that pack their RBDs down more frequently. T372A probably has an incredibly important role in SARS-CoV-2’s transmissibility and therefore pandemic spread, and given it’s likely a related mechanism to what they authors discover and characterize in more detail with their A835D mutation in RsSHC014, the relevance of this mechanism could be emphasized more in relation to pandemic virus emergence.

Reviewer #2: While the use of the ADG2 antibody sufficiently supports the conclusions drawn in the paper regarding epitope accessibility, binding studies performed with a panel of antibodies targeting different regions of the spike protein would be a valuable addition as it would reveal any allosteric effects of the FPPR mutation on other regions of the spike. How the FPPR mutation impacts S2 directed epitopes would be of interest, for example.

This sentence in the discussion “Recent work suggests that, in contrast to SARS-CoV-2, multiple pre-fusion spikes from CoVs of rhinolophid bat and pangolin origin (e.g., BANAL-20-52, BANAL-20-236, RaTG13, Pangolin-CoV), have evolved to favor a more compact ‘closed’ state,” should be modulated/qualified to recognize that the SARS-CoV-2 Omicron variant has been shown to increasingly favor “closed” states.

While the structural conclusions are well supported by the experimental data, it may be worth adding a note that a larger dataset and particle number may have enabled more extensive classification to reveal any small population of up states that may be present. While this would not change any of the conclusions drawn in the paper, to a non-structural, general reader, it would provide an understanding of the methodological caveats in a cryo-EM experiments.

In this sentence “Disruption of RBD-FPPR interaction in the A835D mutant is predicted to free the linker to contribute to the RBD movements associated with spike opening.” The phrase “Disruption of RBD-FPPR interaction” may be confusion as there is no direct interaction between RBD and FPPR. Consider rephrasing. Perhaps “allostery” may be a more suitable word.

In the section titled “A835D does not alter spike expression or VSV incorporation.”, the references to the Supplemental Figures need to be corrected. For example, Figure S2c does not show ELISA data as referred to in the text.

For the Figures reporting infectivity assays, for example Figure 2, the legend should state what reference was used to define 100% infectivity.

Reviewer #3: 11. Please uniform the writing of “SARS-like-CoV” and “SARS-related-CoV”; “RsSHC014” “SHC014”, “SHC014 CoV” and “SHC014-CoV”; “SARS-CoV-1” and “SARS-CoV” and so on, throughout the whole manuscript.

12. Most figures lack clear labels, making data interpretation difficult without figure legends or the main text. The authors should revise the figures for clarity.

13. There are no methods described for assessing spike incorporation into viral particles using ELISA

14. In the text “Despite this, scVSVs bearing the A835D spike were much more infectious than their WT counterparts in Vero cells, recapitulating the phenotypes observed in the rVSV and CoV RDP systems (Fig. S2b). ”，Fig.S2b should be Fig.2c.

15. In the text “In all, we were able to build residues 33-1124, excluding 45–4, 71–78, 94–97, 121–123, 134–155, 167–182, 205–212, 237251, 268–270, 617–621, 665–671, and 812-831.”, please check “45-4”.

16. In the methods “Temperature-Dependent Infection Assays” authors described they used RsACE2, please check.

17. The the method “ADG-2 Neutralization Assay”, authors used DBT-RaACE2 cells, while in the figure 7e legend, Vero cells were used. Please check.

PLOS authors have the option to publish the peer review history of their article (what does this mean?). If published, this will include your full peer review and any attached files.

Reviewer #1: No

Reviewer #2: No

Reviewer #3: No

Figure Files:

Data Requirements:

Reproducibility:

References:

---

## [Editor Report · Decision Letter 1]

29 Oct 2024

Dear Dr. Chandran,

We are pleased to inform you that your manuscript 'Distinct pathways for evolution of enhanced receptor binding and cell entry in SARS-like bat coronaviruses' has been provisionally accepted for publication in PLOS Pathogens (although please see Academic Editor's comments below).

Best regards,

Jesse D Bloom, Ph.D.

Guest Editor

PLOS Pathogens

Alexander Gorbalenya

Section Editor

PLOS Pathogens

Michael Malim

Editor-in-Chief

PLOS Pathogens

orcid.org/0000-0002-7699-2064

The authors have satisfactorily addressed the substantive comments.

I would note that in the revised manuscript, how the citations appear is a bit inconsistent. Sometimes they appear as [1] but *after* the punctuation, sometimes they are superscripted etc. I suspect that all of this will be resolved in copy editing and typesetting, but if there is a version that will be posted for early access (?) then I suggest the authors spend a few minutes fixing that.
---

## [Editor Report · Acceptance letter]

8 Nov 2024

Dear Dr. Chandran,

We are delighted to inform you that your manuscript, "Distinct pathways for evolution of enhanced receptor binding and cell entry in SARS-like bat coronaviruses," has been formally accepted for publication in PLOS Pathogens.

Best regards,

Michael Malim

Editor-in-Chief

PLOS Pathogens

orcid.org/0000-0002-7699-2064